# Significant emissions of DMS and monoterpenes by big leaf Mahogany trees: discovery of a missing DMS source to the atmospheric environment

Lejish Vettikkat[1], Vinayak Sinha[1], Savita Datta[1], Ashish Kumar[1], Haseeb Hakkim[1,], Priya Yadav[1],

5   Baerbel Sinha[1]

[1]Department of Earth and Environmental Sciences, Indian Institute of Science Education and Research Mohali, Sector 81, S. A. S. Nagar, Manauli PO, Punjab, 140306, India

*Correspondence to*: Dr. Vinayak Sinha (vsinha@iisermohali.ac.in)

**Abstract.** Biogenic volatile organic compounds exert a strong influence on regional air quality and climate through their roles in the chemical formation of ozone and fine mode aerosol. Dimethyl sulfide (DMS), in particular, can also impact cloud formation and the radiative budget as it produces sulfate aerosols upon atmospheric oxidation. Recent studies have reported DMS emissions from terrestrial sources , however their magnitudes have been too low to account for the observed ecosystem scale DMS emission fluxes. Big-leaf Mahogany (*Swietenia macrophylla)* is an agro-forestry and natural forest tree known for its good quality timber and listed under the Convention on International Trade in Endangered Species (CITES). It is widely grown in the American and Asian environments (> 2.4 million $km^2$ collectively). Here, we investigated emissions of monoterpenes, isoprene and DMS as well as seasonal carbon assimilation from four big-leaf Mahogany trees in their natural outdoor environment using a dynamic branch cuvette system, high sensitivity proton transfer reaction mass spectrometer and cavity ring down spectrometer. The emissions were characterized in terms of environmental response functions such as temperature, radiation and physiological growth phases including leaf area over the course of four seasons (summer, monsoon, post-monsoon, winter) in 2018-19. We discovered remarkably high emissions of DMS (average in post-monsoon: ~19 ng $g^{-1}$ leaf dry weight $hr^{-1}$) relative to previous known tree DMS emissions, high monoterpenes (average in monsoon: ~15 µg $g^{-1}$ leaf dry weight$hr^{-1}$ which are comparable to oak trees) and low emissions of isoprene. Distinct linear relationships existed in the emissions of all three BVOCs with higher emissions during the reproductive phase (monsoon and post-monsoon seasons) and lower emissions in the vegetative phase (summer and winter seasons) for the same amount of cumulative assimilated carbon. Temperature and PAR dependency of the BVOC emissions enabled formulation of a new parameterization for use in global BVOC emission models. Using the measured seasonal emission fluxes, we provide the first estimates for the global emissions from Mahogany trees which amount to circa 210-320 Gg $yr^{-1}$ for monoterpenes, 370-550 Mg $yr^{-1}$ for DMS and 1700-2600 Mg $yr^{-1}$ for isoprene. Finally, through the results obtained in this study, we have been able to discover and identify Mahogany as one of the missing natural sources of ambient DMS over the Amazon rainforest as well. These new emission findings, indication of seasonal patterns, and estimates will be useful for initiating new studies to further improve the global BVOC terrestrial budget.

## 1 Introduction

Biogenic volatile organic compound (BVOC) emissions contribute to 90% of total annual VOC emissions (Guenther et al., 1995;Fehsenfeld et al., 1992). Of the total BVOC emissions of 1000 Tg $yr^{-1}$ estimated by MEGAN 2.1, terpenoids like isoprene, monoterpenes, and sesquiterpenes contribute about 70% to the total and are emitted majorly in the tropics (Guenther et al., 2012). When mixed with urban air which is typically rich in nitrogen oxides, these highly reactive BVOCs can impact regional air quality significantly by fueling formation of secondary pollutants such as ozone and secondary organic aerosols (SOA) with consequences also for the regional climate (Atkinson and Arey, 2003;Kavouras et al., 1998;Goldstein et al., 2009).

DMS plays a significant role in atmospheric chemistry as it contributes to the formation of ambient sulfate aerosol particles upon atmospheric oxidation. This new particle formation (NPF) can further contribute to direct and indirect radiative forcing by forming cloud condensation nuclei (CCN) (Andreae and Crutzen, 1997). The major biogenic source of dimethyl sulfide (DMS) in the atmosphere are marine phytoplankton (Stefels, 2000;Charlson et al., 1987;Lovelock et al., 1972;Watts, 2000).

However, a recent study from the Amazon rainforest reported high DMS mixing ratios above the forest and concluded that there is a net ecosystem source for DMS (Jardine et al., 2015). Only a few previous studies have shown trees to be potential terrestrial sources of DMS possibly by the uptake of carbonyl sulfide (COS) or from sulfur sources within the tree (Yonemura et al., 2005;Geng and Mu, 2006;Kesselmeier et al., 1993).

Terpenoids play key functional roles in chemical ecology and can be released by plants due to both biotic and abiotic stresses

such as high temperature (Loreto et al., 1998;Sharkey and Singsaas, 1995), intense light (Vickers et al., 2009) and herbivory (Kappers et al., 2011). BVOC emissions are modeled (Guenther et al., 2012) using land use land cover data, temperature, light and other meteorological parameters as key inputs. However, large intra-annual and intra-species variability exist which lead to large uncertainties for annual emission fluxes. In specific instances where the physiological and biochemical pathways responsible for the BVOC emission are also not understood, such as for DMS (Yonemura et al., 2005), it is not even possible

to model the BVOC emissions. Global warming and land use changes further complicate emission flux calculations of BVOCs in models (Peñuelas, 2003;Unger, 2014).

*Swietenia macrophylla* King commonly called the Big-leaf Mahogany is a neotropical tree species which occurs naturally in both the northern hemisphere and southern hemisphere spanning across regions from Mexico (23°N) to the southern Amazon (18°S) and covering an area of circa 150 million hectares (Blundell, 2004). Due to its highly-valued best quality timber,

plantations of this species are also widespread in several parts of South Asia and Southeast Asia (Mayhew et al., 2003). The area under this tree in the American and Asian environments collectively exceeds 2.4 million $km^2$ of land area. This tree species is listed in the Convention on International Trade in Endangered Species (CITES) of Wild Fauna and Flora Appendix II as it faces a threat due to widespread unsustainable logging (Grogan and Barreto, 2005). New silviculture and agroforestry of Mahogany are on an upsurge to sustainably comply with the demand for its timber due to the strict law enforcement, that

prohibits the illegal logging from natural forests which had met the market requirements before the CITES listing (Ward et al., 2008). Varshney et al. 2003 were the first group in India to screen forty tropical Indian trees in terms of their isoprene emission potential, and there now exists a fairly large worldwide database for trees in terms of their isoprene and monoterpene emission potential (http://www.es.lancs.ac.uk/cnhgroup/iso-emissions.pdf). However, to the best of our knowledge, *Swietenia macrophylla* King BVOC emissions have not been investigated previously.

In this study, we investigated emissions of monoterpenes, isoprene and DMS and carbon assimilation from four big-leaf Mahogany trees growing in north India in their natural environment using a dynamic branch cuvette system, a high sensitivity proton transfer reaction mass spectrometer (PTR-MS) and a cavity ring down spectrometer (CRDS). The emissions were characterized in terms of environmental response functions such as temperature, radiation and physiological growth phases including leaf area. While four trees were studied in winter, one of the four trees was also studied over the course of four

seasons (summer, monsoon, post-monsoon, winter) during 2018-19. Using the derived relationships, a new parameterization for use in global BVOC emission models is proposed. Finally, using the seasonal fluxes suggested by the measurements and currently documented natural and planted Mahogany tree cover areas, we provide the first estimates for the global annual emissions of monoterpenes, DMS and isoprene from Mahogany trees.

## 2 Materials and Methods

### 2.1 Sampling, branch cuvette experiments and flux calculation methodology

Table 1 provides a summary of the sampling dates alongwith the average and ambient variability (as standard deviation) of the temperatures and photosynthetically active radiation (PAR) during each of the sampling experiments. A total of four big leaf Mahogany (*Swietenia macrophylla)* trees growing in the natural environment in the north west Indo-Gangetic Plain (30.667 °N, 76.729 °E, 310 m a.s.l.) were sampled using a dynamic branch cuvette sampling system. While sampling and biogenic VOC emission measurements were performed from four Mahogany trees in winter (details in Table 1), the sampling and biogenic VOC emission measurements for three other seasons were from one of the four trees (namely Tree 1 in Table 1) as follows: 2018 summer from 22-24 May ($n\_h$ =52 hours of measurements), 2018 monsoon ($n\_h$ =200 hours of measurements) from 25 September-4 October, 2018 post-monsoon ($n\_h$=163 hours of measurements) from 15-22 November, and 2019 winter from 24-29 January ($n\_h$ =120 hours of measurements). Here, "$n\_h$" refers to the number of hours of measurement in a sample with measurement cycle of a duration of about 3 minutes in summer season and measurement cycle of duration slightly less than a minute during all other seasons. We used the number of hours of measurement as "$n\_h$" to be consistent since we used hourly averaged data for our analysis. Monoterpenes, isoprene, dimethyl sulfide (DMS) were measured using a high sensitivity proton transfer reaction mass spectrometer (PTR-MS; HS Model 11-07HS-088; Ionicon Analytik Gesellschaft, Austria) while carbon dioxide was measured using a cavity ring down spectrometer (CRDS; Model G2508, Picarro, Santa Clara, USA). The same tree was sampled to obtain the inter-seasonal variability. Since observations showed significant DMS emissions we sampled three additional trees, two of which were growing within 10 m of each other and the third of which was growing approximately 250 m away, during wintertime. Tree 1, 2 and 3 were seven-year-old mahogany trees whereas Tree 4 was five years old. All the trees were growing in silty clay soil in outdoor conditions. While two of the three trees were sampled at high temporal resolution continuously in an online manner), offline sampling for collection of whole air samples from the dynamic branch cuvettes was carried out in passivated steel canisters from the distant tree. Below we describe the dynamic branch cuvette system and trace gas measurements.

Polyvinyl fluoride bags (PVF, Tedlar®; 95% transmittance, Dimension: 0.61 m $\times$ 0.91 m, 0.05 mm thickness s. Avg. capacity: 54 l; Jensen Inert Products, Part no. GST002S-2436TJC, USA ) were used as the cuvette material. Previous studies have already discussed its advantages for both analytical and practical purposes (Ortega and Helmig, 2008;Ortega et al., 2008). The bag has one open end and two Jaco fittings (6.3 mm) for inlet and outlet air flow Teflon tubing (0.63 mm, 3.2 mm, 6.3 mm and 9.5 mm I. D., 60-65 m in total with > 95 % length made of 9.5 mm I.D.). The Mahogany branch was equipped with a

temperature (T) and relative humidity (RH) sensor (No: 201403513, HTC easy Log, India) to monitor the cuvette temperature and RH. Ambient meteorological parameters and soil moisture (SM) were also measured using sensors for temperature and RH, PAR and soil moisture (VP-4 RH and T sensor, QSO-S PAR sensor, and GS1 SM sensor, Decagon devices, USA), placed adjacent to the tree. A schematic of the dynamic branch cuvette system can be found in Figure S1. Branches with similar leaf age (ranging from 2-11 months) were selected also ensuring that the cuvette received sunlight throughout the day. Branches with 30-50 leaves of similar leaf age (ranging from 2-11 months) were selected also ensuring that the cuvette received sunlight throughout the day. The cuvette was suspended carefully on the tree branch to minimize the weight stress on the tree and avoid foliage contact within the cuvette. Input air was generated from ambient air using a series of custom built traps containing steel wool, silica gel, and activated charcoal. Measurements of ozone using a portable ozone monitor (PO3M, 2B Technologies, Colorado, US) and the target VOCs in the input air showed that the traps worked quite well with concentrations below detection limit or extremely low values in the input air. A high capacity Teflon VOC pump (Model N145.1.2AT.18, KNF, Germany) was used to ensure a constant flow of air into the cuvette via a mass flow controller (EL-FLOW, Bronkhorst High-Tech Netherlands; stated uncertainty 2%) at 30 l min$^{-1}$. Air from the output port of the cuvette was drawn into the IISER Mohali Atmospheric Chemistry Facility (Sinha et al., 2014) using a second suction pump which drew slightly less than 30 l min$^{-1}$ thereby ensuring a small positive pressure inside the chamber for dynamic and turbulent flow of air through the cuvette. The total inlet length from the cuvette exit to the instruments was 32 m and considering the inner diameter of 9.5 mm and flow rate of ~ 30 l min$^{-1}$, the inlet residence time of air was always less than 10 s for the transfer from the cuvette output to the instruments housed inside the facility. All flows were measured using a NIST calibrated flow meter (BIOS Drycal definer 220, Mesa Labs, US). The input air which served as the background for flux calculations was sampled for all hours of the day in each season by taking measurements 2-3 times a day in each season at different hours of the day, by diverting the air flow such that it bypassed the branch cuvette. After installation of the cuvette, we allowed the branch to acclimatize overnight before starting the measurements to ensure acclimatization/conditioning of leaves to the flows and chamber. This is significantly longer than the steady-state attainment time of circa 5 minutes recommended by Niinemets et al. (2011) but is necessary to prevent measurement artefacts owing to inadvertent physical stress or injuries to the branch immediately after installation. After completion of the measurements, the leaves were destructively harvested from the enclosed branch to measure the total leaf area (m$^2$) inside the cuvette and dried at 60 °C to also measure the leaf dry weight (ldw). Data for the same is available in Table S1.

Whole air was sampled actively for offline measurements in commercially available 6 L passivated SilcoCan air sampling steel canisters (Restek, USA) and then analyzed with PTR-MS and CRDS within 6 hours of sample collection as described in our previous work (Chandra et al., 2017). Briefly, air was sampled into the canisters over a period of 30 minutes at a flow rate of 500 ml min$^{-1}$ to final pressure of 30 psi using a Teflon VOC pump (Model − N86 KT.45.18; KNF, Germany) and mass flow controller (Max. capacity: 500 sccm; Bronkhorst High-Tech; Germany; stated uncertainty 2%).

Emission fluxes for the sum of monoterpenes, isoprene and dimethyl sulfide normalized to leaf area were obtained using Eq. (1) (Sinha et al., 2007;Niinemets et al., 2011)

$$EF_{BVOC} \ (nmol \ m^{-2} \ s^{-1}) = \frac{m_{out,BVOC} - m_{in,BVOC} \ (nmol \ mol^{-1})}{V_m (m^3 \ mol^{-1})} \times \frac{Q \ (m^3 \ s^{-1})}{A \ (m^2)} \qquad (1)$$

where, $m_{out,BVOC} - m_{in,BVOC}$ is the difference in the mixing ratios of the BVOC between output and input air, Q was the flow rate of air passing through the cuvette system in $m^3 \ s^{-1}$, $V_m$ was the molar volume of gas calculated using the cuvette temperature.

The carbon assimilation rate, $A_{net}$ ($\mu mol \ m^{-2} \ s^{-1}$) was calculated using Eq. (2) (Huang et al., 2018)

$$A_{net} (nmol \ m^{-2} \ s^{-1}) = \frac{[CO_{2,in}] - [CO_{2,out}] \ (\mu mol \ mol^{-1})}{V_m \ (m^3 \ mol^{-1})} \times \frac{Q \ (m^3 \ s^{-1})}{A \ (m^2)} \qquad (2)$$

where $[CO_{2,in}] - [CO_{2,out}]$ is the effective $[CO_2]$ taken up by the leaves inside the cuvette. Q and $V_m$ were the same as used in Eq. (1). By comparison with ambient air measurements for the week just before and after the cuvette experiments, it was found that $[CO_{2,in}]$ was equivalent to ambient $[CO_2]$ for the corresponding hour of the day and thus the ambient $CO_2$ values were used as $[CO_{2,in}]$ in Eq. (2).

## 2.2 Isoprene, monoterpene, dimethyl sulfide and carbon dioxide measurements

The output air from the cuvette was sub-sampled into a high-sensitivity proton transfer reaction quadrupole mass spectrometer (PTR-MS; HS Model 11-07HS-088; Ionicon Analytik Gesellschaft, Austria) for the measurements of isoprene, DMS and sum of monoterpenes. The instrument has been previously characterized in detail elsewhere (Sinha et al., 2014;Chandra et al., 2017;Kumar et al., 2018). In this technique, most analyte molecules having a proton affinity greater than water vapour (165 kcal $mol^{-1}$) undergo soft chemical ionization with reagent hydronium ions ($H_3O^+$) inside a drift tube to form protonated organic ions which are typically detected at mass to charge ratios (m/z) = molecular ion + 1. The product ions are then separated using a quadrupole mass analyzer and detected using a secondary electron multiplier. Measurements were conducted in mass scan mode during summer season and the ion selective in subsequent seasons typically with a dwell time of 1s at each VOC specific m/z channel. Compound-specific sensitivities (ncps $ppb^{-1}$) were determined using calibration experiments involving dynamic dilution of a VOC gas standard (Apel–Riemer Environmental, Inc., Colorado, USA; containing thirteen VOCs at circa 500 ppb; details provided in Table S2) on 4 May 2018, 4 October 2018, 14 November 2018 and 22 January 2019. The pre-mixed VOC gas standard (Apel–Riemer Environmental,Inc., Colorado, USA) contained 495 ppb of dimethyl sulfide (detected at m/z 63), 483 ppb of isoprene (detected at m/z 69) and 494 ppb of the monoterpene α-pinene (detected at m/z 137 and m/z 81 after fragmentation). The stated accuracy of the VOC standard was 5% for all these compounds and as stated in the manufacturer's certificate several of the compounds remain stable even beyond the one-year period mentioned in the certificate. We also verified the same for DMS, isoprene and alpha-pinene by comparison with newer VOC gas standards for which the certificate was still valid and is a standard practice in our laboratory to keep track of any changed concentrations inside the VOC standard after the expiry date (see for e.g. Table S1 of Sinha et al., 2014). The gas standard was dynamically diluted with VOC free-zero air generated using a Gas Calibration Unit (GCU-s v2.1, Ionimed Analytik, Innsbruck, Austria). The flows of both the standard gas and zero air mass flow controllers were measured independently before and after the calibration experiments

using a NIST calibrated flow meter (BIOS Drycal definer 220, Mesa Labs, US). Figure S2 presents data from two calibration experiments conducted on 4 May 2018 and 4 October 2018, that show there was very little drift in sensitivity of the PTR-QMS for the three compounds (DMS < 3.8%; isoprene < 4.1 % and alpha-pinene < 6.1 %) even over a period spanning 5 months. The uncertainties were calculated using the root mean square propagation of individual uncertainties including the instrumental

precision error, 5% accuracy error inherent in the VOC gas standard and 2% precision error of the MFCs as explained in Sinha et al. 2014. For offline measurements, the standard deviation associated with the average value obtained for each canister measurement already included the instrumental precision error and mass flow controller precision errors. The procedure for calculation of the uncertainties in mixing ratios and emission fluxes has been detailed in the supplement. Table S3 lists the sensitivity factor, limit of detection, instrumental uncertainty and total measurement uncertainty for isoprene, DMS and sum

of monoterpenes. The total measurement uncertainty was found to be less than equal to 13 % for isoprene, DMS and sum of monoterpenes also accounting for the instrumental background (determined by sampling VOC free air) at these m/z ratios. Extensive reviews (de Gouw and Warneke, 2007;Yuan et al., 2017) of previous PTR-MS studies including inter-comparisons with other more specific techniques as well as more recent validation experiments for DMS detection (Jardine et al., 2015) have demonstrated that under standard PTR-MS operational conditions ranging from 130-135 Td), isoprene and dimethyl

sulfide can be detected at m/z 69 and m/z 63, respectively without any significant fragmentation and that as monoterpenes fragment their quantification can be accomplished by taking the sum of the major ions formed, namely m/z 81 and m/z 137 (Lindinger and Jordan, 1998;Tani et al., 2003). We, therefore, operated the instrument under standard operating conditions of drift tube pressure of 2.2 mbar, drift voltage of 600 V and temperature of 60 °C which yields a Townsend ratio of 135 Td. It resulted in a steady and very high primary ion count (1.3-2.5 x $10^7$ counts per second (cps) $H_3O^+$) and low water cluster

(average abundance < 4.1% of primary ion). In the next few paragraphs, we provide a detailed description about the steps we took to account for potential interferences concerning identification of DMS, isoprene and the sum of monoterpenes using our PTR-QMS.

When we commenced the first set of plant cuvette measurements in summer we undertook mass scans for the input air and output air into the branch cuvette over the entire mass range of (m/z 21- m/z 210) during the experiments. We found that in

comparison to the ambient air, the mass scans contained very few peaks and the spectra was remarkably simple (see Fig S3). The results of these mass scans formed the basis for our choice of what masses to monitor in subsequent plant chamber experiments in other seasons from the same tree. Despite the simple spectra obtained in our mass scan results during summer, for subsequent experiments conducted in the selected ion monitoring (SIM) mode in other seasons, we still monitored 60 m/z channels of interest keeping in mind the PTR-MS literature for BVOC emissions, major atmospheric VOCs, and abundant

ions formed generally due to the ion chemistry in the PTR-MS drift tube, which include impurity ions such as m/z 30 ($NO^+$), m/z 32 ($O_2^+$) etc.. . The list of 60 also included m/z 42, m/z 43, m/z 44, m/z 45, m/z 46, m/z 47, m/z 48, m/z 49, m/z 55, m/z 57, m/z 58, m/z 59, m/z 60, m/z 61, m/z 63, m/z 65, m/z 67, m/z 68, m/z 69, m/z 70, m/z 71, m/z 72, m/z 73, m/z 74, m/z 75, m/z 79, m/z 81, m/z 83, m/z 85, m/z 87, m/z 88, m/z 89, m/z 91, m/z 93, m/z 95, m/z 97, m/z 99, m/z 100, m/z 101, m/z 105,

m/z 107, m/z 109, m/z 119, m/z 121, m/z 123, m/z 129, m/z 135, m/z 137, m/z 149, and m/z 205. This enabled us to examine also scope for any potential new interferences due to fragmentation/clustering effects and/or new emissions.

To rule out the possibility of any higher compounds fragmenting and contributing to the m/z 63 signal in our dataset, we
undertook correlation of all other monitored m/z at which measurable signal was observed with the m/z 63, but found no significant correlation ($r^2 \leq 0.2$) with any of them, which suggested that fragmentation of a larger volatile detected at higher mass to charge ratio was likely not responsible for the observed m/z 63 signal. In particular, concerning the potential for other sulphur containing compounds such as dimethyl disulfide ($CH_3SSCH_3$, DMDS), and dimethyl trisulfide ($CH_3SSSCH_3$, DMTS), fragmenting and contributing to the m/z 63 signal, we would like to note that the parent ions of these compounds
would be detected at m/z 95 and m/z 127. As mentioned above we did monitor m/z 95 in all the seasons but didn't monitor m/z 127 in the experiments after summer season as we didn't see any signal at this m/z in the output air of the cuvette. We also could not find any previous report suggesting the possibility of these compounds fragmenting to m/z 63 under standard operating conditions of the PTR-MS such as 135 Td at which we operated our PTR-QMS. On the contrary, a recent relevant study conducted using both GC-MS and PTR-TOF-MS (under similar range of operating conditions; 120-140 Td) for
organosulfur compounds which included these compounds (Perraud et al., 2016), showed that dimethyl disulfide ($CH_3SSCH_3$, DMDS), and dimethyl trisulfide ($CH_3SSSCH_3$, DMTS) do not fragment and contribute to the m/z 63 channel, at which DMS is detected. Our own mass scans and correlation analyses are also consistent with these findings and so we were able to rule out the possibility of such higher compounds fragmenting and contributing to the m/z 63 signal in our dataset.

The issue of hydration of protonated acetaldehyde which can form the following ion: $H^+ (CH_3CHO) H_2O$ (which has m/z 63) and therefore could contribute to the m/z 63 attributed to DMS required careful attention. This issue was first pointed out in the review by de Gouw and Warneke 2007 and further addressed adequately in the work by Jardine et al. 2015. The interference from this ion can be significant when both the hydrated hydronium ion and acetaldehyde concentrations are high leading to appreciable formation of $H^+ (CH_3CHO) H_2O$ in the drift tube from reactions of the $H^+ (CH_3CHO)$ with $(H_3O^+H_2O)$ ion. As
shown in the work of Jardine et al. 2015, if the abundance of the hydrated hydronium ion ($H_3O^+H_2O$) is therefore kept to just a few percent of the primary reagent ion namely the $H_3O+$ ion (circa 4 %), then at mixing ratios of less than 19 ppb acetaldehyde that occur in most ambient environments and well ventilated cuvette systems, this interference has been shown to be negligible (see for example results reported in the paper by Jardine et al. 2015, where even at acetaldehyde mixing ratios as high as 19 ppb, there was no measurable change in the m/z 63 ion signal). We therefore took the above precaution of operating under
high Townsend ratios (~135 Td) in the drift tube to minimize conditions that favour formation of clusters ions by enhancing kinetic energy of the reagent ions. During all our experiments, acetaldehyde mixing ratios were below 12 ppb and under our operating conditions (135 Td), the average $H_2O H_3O^+$ to $H_3O^+$ ratio was only 4.12 % for the entire dataset which is comparable to the 4% or lower abundance during experiments conducted by Jardine et al., 2015. Our dataset was further carefully examined for indications of this potential interference biasing the measured m/z 63 attribution to DMS. For this we plotted the 4 min

averaged temporal resolution primary data for m/z 63 ion against the corresponding co-measured 4 min averaged temporal resolution primary m/z 45 ion data for all the seasons. The results are shown in Figure S4 (a), where it can be seen that there was no significant correlation between the two (r = 0.22) and even at high m/z 45 mixing ratios of 10 ppb, low m/z 63 mixing ratios of 0.2 ppb occurred frequently, which would not have been the case if the m/z 63 originated primarily from the acetaldehyde hydrated water ion cluster. Therefore, in view of the above, just like Jardine et al. 2015, we are confident that the potential interference of acetaldehyde on the DMS measurements was absent/negligible. The measured DMS signals were generally too low to clearly observe the shoulder isotopic peaks originating from the abundance of the 13C, 33S and 34S isotopes. However, during the summer time, when the PTR-MS was operated in the mass scan mode there were periods wherein the DMS signal (m/z 63) was sufficiently high (~0.5 ppb) to observe the isotopic peaks at m/z 64 and m/z 65 (e.g. during noontime on 22.05.2019). Figure S4 (b) shows the 30-minute averaged mass spectra of m/z 63, 64 and 65 during one such occasion. Based on the natural isotopic distribution of 13C, 33S and 34S, one would expect approximately 3.0 % and 4.5 % signal from m/z 63 to land at m/z 64 and 65, respectively and the data in Fig S4 (b) showing the signals observed at m/z 64 and m/z 65 are consistent with the same. These peaks were also comparable with the mass spectra obtained while calibrating the PTR-MS at DMS mixing ratios of 0.5 ppb. Hence these additional supporting evidence from the shoulder isotopic peaks in combination with previous reports in the literature concerning detection of DMS with PTR-MS provide clear evidence that the signal at m/z 63 observed with the PTR-MS in our dataset can be attributed majorly to DMS.

The attribution of isoprene to m/z 69 also requires careful attention and consideration of known interferences from isobaric compounds and fragments of higher ions. As mentioned in the excellent review by Yuan et al. 2017, several compounds can present substantial interferences in various environments, such as furan in biomass-burning plumes, cycloalkanes in urban environments and oil/gas regions, 2-methyl-3-buten-2-ol (MBO) in pine forests, and methylbutanals and 1-penten-3-ol from leaf-wound compounds. We examine one by one each of these possible interferences for the isoprene measurements reported in our dataset. Firstly, we note that many of the potential interferences that can affect the m/z 69 signal while sampling ambient air influenced by mixed combustion and biogenic sources are not relevant for our experimental set up as the output air from the branch cuvettes (after subtracting input air) is exclusively influenced by only biogenic emissions. Concerning the other biogenic emissions that could still be responsible for contributing to the m/z 69 signal measured by the PTR-MS, we could identify isoprene as the main contributor based on isoprene measurements in the output air of the cuvette obtained using a Thermal Desorption- Gas Chromatography-Flame Ionization Detector (TD-GC-FID) system simultaneously. Even though the data was semi-quantitative due to suspected transfer losses noted subsequently within the GC system, they adequately prove that the air from the branch cuvette contained isoprene. Details of the chromatographic detection of isoprene (Figure S5) time series (Figure S6) and its correlation (r = 1) (Figure S7) with the measured m/z 69 signal in the PTR-QMS for the monsoon season are provided in the supplement. When combined with the observed diurnal variability of the m/z 69 PTR-MS signal with PAR and temperature, and these additional observations using the TD-GC-FID, it is clear that no other known compound other than isoprene could satisfy all the above criteria. Hence m/s 69 was confidently attributed to isoprene.

The sum of monoterpenes can be detected using the PTR-QMS technique collectively at m/z 81 (major fragment ion) and m/z 137 with the typical fragmentation ratio ranging from 60-65 % at m/z 81 and 40-35% at m/z 137, depending on the structure of the major monoterpenes that contribute to the sum of the monoterpenes. For alpha-pinene, during our calibration experiments we found that at under the conditions we operated our PTR-MS (~135 Td), 65% of the signal landed at m/z 81 and 35 % at m/z 137. As we cannot rule that the major monoterpene emitted from Mahogany trees is not alpha-pinene, we chose to take the sum of m/z 81 and m/z 137 signals for quantifying the monoterpenes, instead of only m/z 137. Of course while doing so, one has to check that other isobaric ions due to compounds that are not monoterpenes do not contribute majorly to m/z 81. For this we examined the correlation between observed m81 and m137 signals from the plant chamber output air for all seasons. The results showed that m/z 81 originating from some ion other than m137, was unlikely (r=1 between m/z 137 and m/z 81) for all seasons (see in Figure S8). The near perfect correlation also suggests that the composition of the monoterpenes was not different from one season to another because if different monoterpenes with different fragmentation ratios between m/z 81 and m/z 137 were emitted, all the points would not lie on the same line. The isotopic shoulder peaks (m/z 82 and m/z 138 due to natural C-13 abundance) shown in the mass spectra (Figure S3) were also consistent with ions originating from monoterpenes. Hence we could attribute the observed m/z 81 and m/z 137 ions to sum of monoterpenes emitted by Mahogany.

Carbon dioxide measurements were performed by sub-sampling air from the cuvette into a cavity ring down spectrometer (CRDS; Model G2508, Picarro, Santa Clara, USA) which has been described in previous works from our group (Chandra et al., 2017). The overall uncertainty for measurements of $CO_2$ was below 4%. The instrument was calibrated by dynamic dilution of a gas standard mixture (1998 ppm $CO_2$ in Nitrogen traceable to NIST, USA, 2 % uncertainty; Sigma gases, India) on 8 June 2018, 26 October 2018 and 24 January 2019.

### 2.3 Statistical analysis of the dataset

The high temporal resolution data of the BVOCs, CO2 and environmental parameters like temperature and light intensities were averaged to hourly values and used for analysis and interpretation of results. The Kruskal-Wallis test using the PAleontological Statistics (PAST) Version 3.25 software was performed to check if temperature, light intensities of the different season and the corresponding BVOC emissions were significantly different since it is a robust way to compare two or more independent samples of different sizes that are not normally distributed. The correlations of dimethyl sulfide, isoprene and monoterpene emissions to variations in temperature, light and cumulative $CO_2$ assimilation were assessed by Pearson's correlation. The effects of temperature and light on BVOC emission flux was modelled, and all other graphing and statistical analyses were performed using IGOR 6.37.

# 3 Results and discussion

## 3.1 Emission of BVOCs from Mahogany including light and temperature dependency

Figure 1 shows the average wintertime emission fluxes and variability (as standard deviation) from trees 2, 3 and 4 shown in comparison to the average flux and variability of tree 1. Earlier in Table 1, a summary of the sampling, PAR and temperature
data for these experiments has already been provided. It can be observed that the observed hourly emission fluxes from Tree 1 (which was also sampled in three other seasons as mentioned in Section 2.1) were always within the observed one sigma variability of the emission fluxes for monoterpenes and isoprene obtained from Trees 2, 3 and 4. For DMS, the observed daytime emission fluxes from Tree 1 were at times lower than the 1 sigma variability range of the DMS flux observed from Trees 2, 3 and 4, and at the lower end of the observed emission fluxes from the other trees. This implies that the DMS fluxes
obtained using Tree 1 do not overestimate the DMS emission fluxes for *Swietenia macrophylla*. Overall, based on comparison with three other replicate trees of Mahogany (trees 2, 3 and 4) for the wintertime data, one can surmise that there is no evidence of Tree1' s emission profile and emission fluxes being anomalous.

Figure 2 shows the measured hourly averaged emission flux from big leaf Mahogany normalized to leaf area for the sum of monoterpenes and isoprene (top panel), DMS (middle panel 1), photosynthetically active radiation, along with the temperature
(middle panel 2) and relative humidity (bottom panel) during summer, monsoon, post-monsoon and winter. Clear diurnal variation was observed in the emission profiles of all three compounds in all seasons with emissions reducing to zero/negligible emission fluxes in all seasons at night when PAR was zero. Average temperatures were highest in summer (~35±5 °C), followed by the monsoon (~30±8 °C), post-monsoon (~21±7 °C) and winter season (~13.5 ±7°C). Peak hourly PAR ranged from 0-1200 $\mu mol\ m^{-2}s^{-1}$ in all seasons except the post-monsoon where maximum hourly values remained below 900 $\mu mol\ m^{-2}s^{-1}$
on all days of sampling. The Kruskal-Wallis test results revealed that the temperature ($p<0.01$) and light intensities ($p<0.01$) in different seasons, as well as the corresponding BVOC emissions ($p<0.01$) are significantly different at 99 % confidence interval or more. Thus, emission fluxes obtained in this study covered a fairly large range of ambient temperature and light conditions. The summertime measurements were performed only for 52 hours, but a comparison of the meteorological data for this period with the meteorological data before and after the sampling period showed that the sampling was carried out
under conditions characteristic of the typical summer time conditions (low daytime RH and high temperature and PAR). Winter was associated with the lowest BVOC emission fluxes for monoterpenes and isoprene (avg for both $< 0.05$ nmol $m^{-2}\ s^{-1}$) as well as DMS (avg 1.7 pmol $m^{-2}\ s^{-1}$), even though peak PAR values in winter were comparable to monsoon and summer. Thus, temperature was a major driver for emissions of all three compounds in the winter season. Average monoterpene emission fluxes were highest in the monsoon season (2.3 nmol $m^{-2}\ s^{-1}$) followed by the post-monsoon (~1.7 nmol $m^{-2}\ s^{-1}$) and summer
season (~1.5 nmol $m^{-2}\ s^{-1}$), revealing that Mahogany is a high monoterpene emitter comparable to the highest monoterpene emitting trees in the world such as oaks (http://www.es.lancs.ac.uk/cnhgroup/iso-emissions.pdf) and actively so throughout the year. Average DMS emission fluxes were highest in summer season (~8.2 pmol $m^{-2}\ s^{-1}$), closely followed by post-monsoon

season (~7.1 pmol m$^{-2}$ s$^{-1}$) and monsoon season (~5.3 pmol m$^{-2}$ s$^{-1}$), with lowest emissions during the winter season (~1.8 pmol m$^{-2}$ s$^{-1}$). As most previous studies in the literature have reported emission fluxes of different tree species normalized to the leaf dry weight per hour in Table 2 we provide the average emission fluxes for each season in these units as well. In comparison, isoprene emission fluxes were significantly lower with average emission fluxes of only 0.03 nmol m$^{-2}$ s$^{-1}$ being observed during summer, monsoon and post-monsoon. The time series of BVOC mixing ratios in output air of the cuvette alongwith the background mixing ratios in input air are shown in Figure S9 for Tree 1 and Figure S11 for Trees 2,3 and 4, Figure S10 shows the wintertime BVOC emission fluxes for Trees 2,3 and 4 along with PAR and temperature. (expressed in nanomols or picomols per leaf area per second). The emission profiles of monoterpenes and isoprene co-varied and correlated strongly in all seasons (r$^2$≥ 0.8 with r$^2$≥0.9 during summer and monsoon). This indicates that their emissions arise from common pathways in Mahogany and that fresh photosynthetically fixed carbon may be more important than emissions from stored pools (Monson et al., 1995). DMS emissions also correlated with the terpene emissions in all seasons except winter (r$^2$ = 0.2) but were much weaker (0.4 ≤ r$^2$ ≤ 0.5).

Whereas databases now exist concerning isoprene and monoterpene emission potential of trees, and also many studies have shown that monoterpene and isoprene emissions depend on the plant functional type, PAR availability, temperature and to a lesser extent soil moisture (Kesselmeier and Staudt, 1999;Guenther et al., 1996)  (http://www.es.lancs.ac.uk/cnhgroup/iso-emissions.pdf), there are very few studies in the literature reporting DMS emissions from terrestrial plants and ecosystems (Kesselmeier et al., 1993;Yonemura et al., 2005;Geng and Mu, 2006), with even less known about the factors that control DMS emissions (Jardine et al., 2015). Hourly averaged DMS emission flux from Mahogany was found to vary between a maximum of 15.7 pmol m$^{-2}$ s$^{-1}$ in winter to 48.2 pmol m$^{-2}$ s$^{-1}$ in the post-monsoon seasons and were much higher than the maximum flux of 26 pmol m$^{-2}$ s$^{-1}$ observed from Hibiscus sp (Yonemura et al., 2005) for the DMS branch emission measurements made from seven tropical plant species (max ~6 pmol m$^{-2}$ s$^{-1}$) within a large, enclosed rainforest mesocosm in Arizona, USA (Jardine et al., 2015) and the Geng and Mu (2006) study in China ( max ~2 pmol m$^{-2}$ s$^{-1}$). We note that in all these previous studies the range of temperature and PAR covered while measuring the DMS were significantly lower, with the temperature never exceeding 30 °C and PAR lower than 140 µmol m$^{-2}$s$^{-1}$ in the Jardine et al. 2015 study and less than 500 µmol m$^{-2}$s$^{-1}$ in the Yonemura et al., 2005 study, respectively.

To investigate the factors driving the emissions of monoterpenes, isoprene, and DMS in different seasons from Mahogany, we examined the relationship between the cumulative BVOC emission flux of these compounds with respect to the cumulative CO$_2$ assimilation flux (A$_{net}$) starting from the sunrise of each day. Cumulative emission fluxes were calculated for every hour of the day and accumulated from sunrise until that hour. This is helpful as A$_{net}$ is a good proxy for the rate of photosynthesis and a recent $^{13}$C-pulsed labeling study has shown that newly assimilated carbon can be emitted as monoterpenes within one hour (Huang et al., 2018). Further, depending on whether the tree's growth is in the reproductive or vegetative phase (Huijser & Schmid 2011), the assimilated carbon can be allocated differently impacting the emitted BVOC flux. For example, one could expect that in the constitutive growth phase, emissions of BVOCs would be lower whereas, in the reproductive phase, when flowering and fruiting occur, due to the important functional roles BVOCs play in attracting pollinators and for plant

defence, there would be increased emissions of BVOCs (Peñuelas, 2003). Mahogany is known to bear fruits during the monsoon season (Gullison et al., 1996) and trees emit odorous compounds like terpenes for defence purposes especially against herbivores and abiotic stresses like high-intensity light, temperature. Hence the enhanced emission of BVOCs during the monsoon and post-monsoon seasons is likely due to these reasons. This diversion of the carbon allocation for such purposes can decrease growth by diverting photosynthates from the production of vegetative structures (Herms and Mattson, 1992). Henceforth, the two distinct phases are referred to as the vegetative growth phase when the carbon allocation to BVOC synthesis is low and reproductive growth phase, when the carbon allocation by the tree to synthesize BVOCs is high. The results are shown in Figure 3(a) for monoterpenes, isoprene, and DMS. Distinct linear relationships were observed for the emissions of all three BVOCs with higher emissions during the reproductive phase (monsoon and post-monsoon seasons) and lower emissions in the vegetative phase (summer and winter seasons) for the same amount of cumulative assimilated carbon. It is interesting to note that DMS flux also shows this pattern in the two phases which suggests that DMS emission could be linked to these functional roles as well, in addition to being dependent upon the uptake of COS, the latter of which has been previously reported to be similar to uptake of carbon dioxide during photosynthesis (Jardine et al., 2015).

Global BVOC emission models such as MEGAN - Model of Emissions of Gases and Aerosols from Nature (Guenther et al., 2012) use PAR and ambient temperature dependence of major plant functional types to calculate BVOC emissions. Thus, it is meaningful to examine if one can obtain a parameterization of the monoterpene, isoprene, and DMS flux from big leaf Mahogany trees in terms of PAR and temperature. Figure 3(b) shows 3-D surface plots illustrating the dependence of BVOC emission flux as a function of instantaneous chamber temperature and PAR in the vegetative growth phase. In the vegetative phase, terpenes varied exponentially with respect to the two meteorological drivers. It is also evident that DMS has a strong dependence on temperature, but not on PAR. DMS peaked during high temperatures even when PAR was only 200 μmol m$^{-2}$s$^{-1}$. However, the dependence of DMS flux on temperature is not always followed possibly because the DMS flux is dependent upon the uptake of COS or on the internal sulfur content. From Figure 3 (b) it also appears that the temperature has no effect on the DMS emission flux at low PAR (< 400 μmol m$^{-2}$ s$^{-1}$). We constructed best bivariate fit functions by expressing the emission flux as an exponential function of both temperature and PAR for the vegetative growth phase and as a linear function of PAR, and an exponential function of temperature in the reproductive growth phase to better formulate the dependence of the BVOC emissions on these meteorological parameters.

Table 3 shows the fit functions and their coefficients for BVOC flux parameterizations as a function of PAR and temperature in both the reproductive and vegetative phases of Mahogany. The temperature dependent coefficient in the reproductive growth phase (c) is much lower than the temperature dependent coefficient in the vegetative growth phase (d). This implies that during the reproductive phase plant emits higher BVOCs with less temperature increment than during the vegetative phase and is in agreement with our earlier observation regarding the higher carbon allocation for the BVOC synthesis and emission during the reproductive growth phase.

Figure 3(c) shows the modeled BVOC emission fluxes and measured BVOC emission fluxes for all the seasons. The observed temperature and PAR data during the experiments were used to calculate the modeled flux using the bivariate fit function for

the two growth phases. We found that the measured flux can be predicted only if both the functions are used to calculate the modeled flux of the respective phase. Modeled DMS showed deviations from measured flux which may be attributed to irregularity in the dependence on high temperature but currently in the absence of knowledge concerning the exact pathways responsible for DMS emission, the reasons remain unclear. Still, the finding that vegetative growth and reproductive growth

phases require different modeling functions, point to the need for considering the phenological cycle changes of plants in annual emissions as these can result in a significant increase or decrease in the modeled BVOC emissions from similar vegetation. These parameterizations provide a way to simulate Mahogany emissions even in global BVOC emission models that already use the PAR and temperature data for simulation of BVOC emissions.

### 3.3 Estimates of global annual emissions of monoterpene, isoprene, and DMS from Mahogany

Table 4 shows the distribution of Mahogany in natural forests and in plantations in terms of ground area, density, leaf area and calculated annual emission fluxes of monoterpenes, isoprene, and DMS for several countries, based on the documented area under Mahogany tree cover. First, the Mahogany tree cover was estimated using the available data regarding the natural forest and plantation cover in different countries around the globe (Blundell, 2004;Lugo et al., 2003;Mohandas, 2000). Forest cover was multiplied by the density of Mahogany trees reported in those countries (Gullison et al., 1996;Lugo et al., 2003;Gillies et

al., 1999;Grogan et al., 2008;Kammesheidt et al., 2001) to estimate the total number of Mahogany trees in the world. The total crown size was calculated using the equation provided by a pioneering study by Gullison et al. (1996), assuming the median diameter at breast height (DBH) to be 80 cm  in forests. This was multiplied by leaf area index (LAI) (Jhou et al., 2017) to obtain the leaf area. For plantations where density was unavailable, the plantation area was multiplied by LAI to obtain the leaf area. The annual emission fluxes were calculated assuming six months of reproductive and vegetative phase each, and the

average measured emission fluxes normalized to leaf area obtained in our study for each of these phases. The Table lists both natural and plantation area cover for Mahogany, and it can be seen that Brazil and several other regions in South America stand out with Brazil alone having more than 1.4 million square kilometres of Mahogany tree cover. In terms of large planted tree areas, several regions in Asia such as Indonesia and the Philippines stand out. We would like to point out that this estimate is based on the current available information but there may be some underestimation as there are areas where cultivation of

Mahogany trees is known to occur (e.g. Jim Corbett national park in India), for which, however, accurate Mahogany biomass estimates are not yet available and which hence were not included in Table 4.The list is nonetheless useful to identify regions where the influence of DMS and monoterpene emissions from Mahogany are important to consider for regional air quality and climate, through aerosol and oxidant chemistry feedbacks. In this context, recent ecosystem scale DMS emissions reported over the rainforest in South America (Jardine et al., 2015) could indeed be partially explained by the contribution of DMS

emissions from Mahogany growing in the rainforest and surrounding areas. Further, high monoterpene and DMS emissions from Mahogany would also contribute through the formation of aerosol particles. Our estimates indicate global yearly DMS emissions of 370-550 Mg from Mahogany alone. Further, as the cultivation of Mahogany is gaining popularity in southern

Asia and are already significant in Indonesia and Fiji due to huge plantations, focused studies on the regional impact of these plantations through BVOC feedbacks to climate and air quality are warranted. Based on results obtained in this study, *Swietenia macrophylla* is estimated to also emit 210-320 Gg yr$^{-1}$ of monoterpenes globally, with most of the emissions concentrated in specific regions of South America, Asia, and North America. The total isoprene emission flux does not seem to be of much consequence for the global budget of isoprene as it amounted to only 2600 Mg yr$^{-1}$ but could still be of significance regionally as a dominant isoprene source, and require further investigations.

**4 Conclusions**

In this study, BVOC emissions of monoterpenes, isoprene and DMS were determined in four different seasons at branch level from *Swietenia macrophylla* King (also called big leaf Mahogany) growing in their natural environment in India. The emissions were characterized in terms of environmental response functions such as temperature, radiation and physiological growth phases. Branch level measurements revealed remarkably high emissions of DMS (average in post-monsoon: ~19 ngg$^{-1}$ leaf dry weight hr$^{-1}$) relative to previous known tree DMS emissions, high monoterpenes (average in monsoon: ~15 µgg$^{-1}$ leaf dry weight hr$^{-1}$ which are comparable to high emitters such as oak trees) and low emissions of isoprene ($< 0.09$ µgg$^{-1}$ leaf dry weight hr$^{-1}$). Distinct linear relationships were observed between cumulative BVOC emissions and the cumulative assimilated carbon with higher emissions during the reproductive phase (monsoon and post-monsoon seasons) and lower emissions in the vegetative phase (summer and winter seasons) for the same amount of cumulative assimilated carbon. Temperature and PAR dependency of the BVOC emissions enabled formulation of a new parametrization that can be employed in global BVOC emission models. Using the seasonal fluxes suggested by the measurements , we provide the first global emission estimates from Mahogany trees of circa 210-320 Gg yr$^{-1}$ for monoterpenes, 370-550 Mg yr$^{-1}$ for DMS and 1700-2600 Mg yr$^{-1}$ for isoprene.

While several novel insights have been obtained from this study such as discovery of a new terrestrial source with high emissions for monoterpenes and DMS relative to other known terrestrial sources, one limitation has been the lack of data from replicates for three of the four seasons. Based on comparison with three other replicate trees of Mahogany (Trees 2, 3 and 4) for the wintertime data, one can surmise that there is no evidence of Tree1's emission profile and emission fluxes being anomalous and hence considering the paucity of what is known about DMS seasonal emissions from trees (this study to the best of our knowledge contains first such information on seasonal emission tendencies), the insights about seasonality of Mahogany emissions obtained in this study are also valuable. We acknowledge, however that data from more replicates would be better to characterize the intra-species and seasonal emissions variability better and should be addressed in future studies, and the reported seasonal values in this study need to be treated with caution as seasonal changes of VOCs could be strongly tree-specific especially when the emissions are controlled by enzymatic processes.

Since Mahogany has a large vegetation cover in the Mesoamerican forests and is gaining popularity in South Asia due to its economic significance, large-scale emissions through land use land cover changes from this species could have a significant impact on local and regional atmospheric chemistry. Finally, through the results obtained in this study, we have been able to discover and identify Mahogany as one of the missing natural sources of ambient DMS over the Amazon rainforest as well.

5    These new emission findings, indication of seasonal patterns, and estimates will be useful for initiating new studies to further improve the global BVOC terrestrial budget.

**Data availability.** Data is available from the corresponding author upon request

**Author contributions.** V.S. and B.S. conceived and designed the study.  L.V. carried out this work as part of his MS thesis under the supervision of V.S. L.V. performed PTR-MS measurements with help from H.H. and carried out preliminary analysis and wrote the first draft. V.S. revised the paper and carried out advanced analyses and interpretation of the data and supervised all experimental aspects of the work. S.D., A.K., H.H. and P.Y. contributed to the plant cuvette sampling experiments and CRDS measurements. B.S. commented on the revised draft and helped with compilation of Table 4.

**Competing interests.** The authors have no competing interests to declare.

**Acknowledgements.** We acknowledge the IISER Mohali Atmospheric Chemistry facility for data and the Ministry of Human Resource Development (MHRD), India for funding the facility. L.V. and P.Y, A.K., H.H. acknowledge IISER Mohali for MS fellowships and Institute PhD fellowships while S.D. acknowledges UGC for Ph.D fellowship. This work was carried out under the National Mission on Strategic knowledge for Climate Change (NMSKCC) MRDP Program of the Department of Science and Technology, India vide grant (SPLICE) DST/CCP/MRDP/100/2017(G)**.** We acknowledge EGU for waiver of the APC through the EGU 2019 OSPP award to L.V and the two anonymous reviewers for their helpful suggestions and insightful comments which helped to improve the discussion paper.

.

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

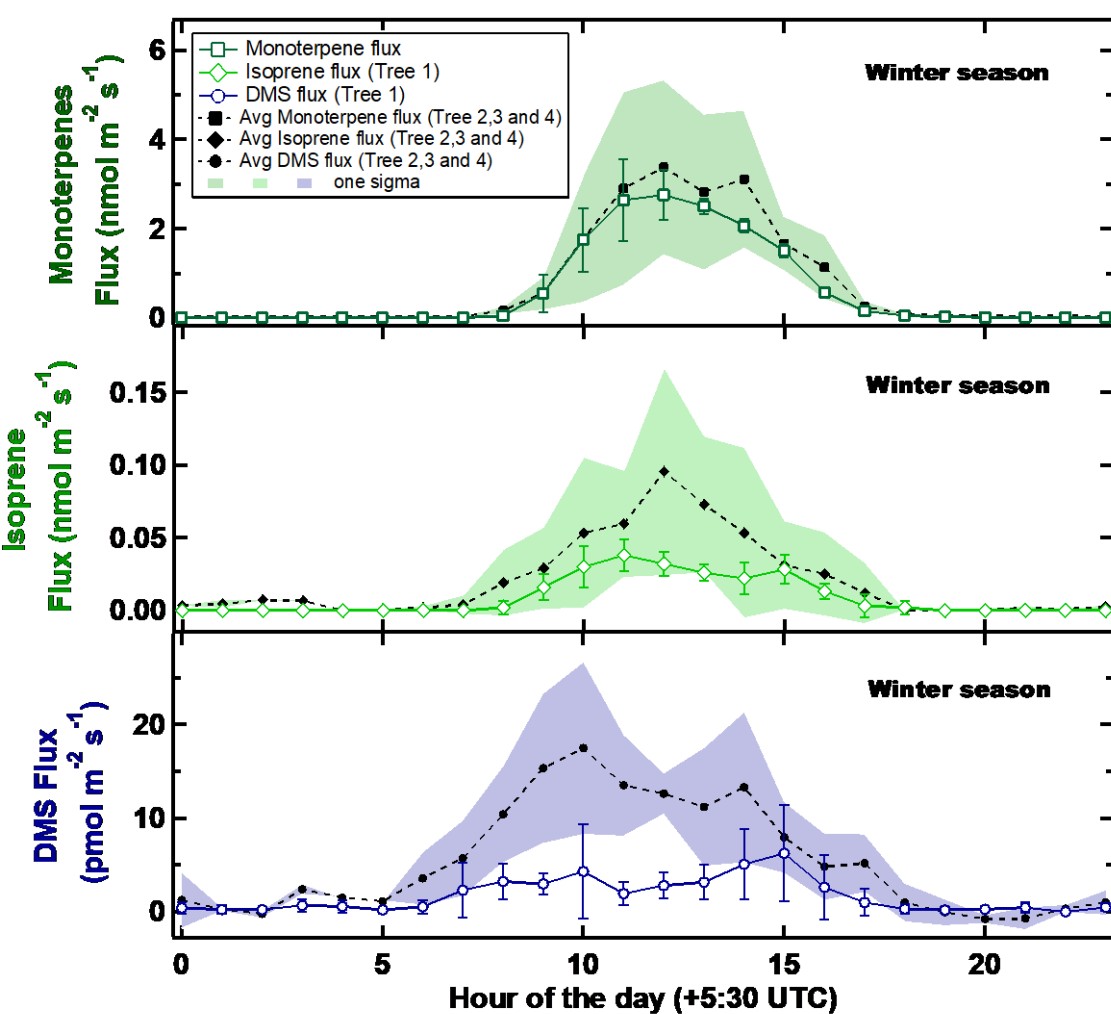

**Figure 1.** Average wintertime emission fluxes and variability (as standard deviation) for Trees 2, 3 and 4 shown in comparison to average flux and variability of Tree 1

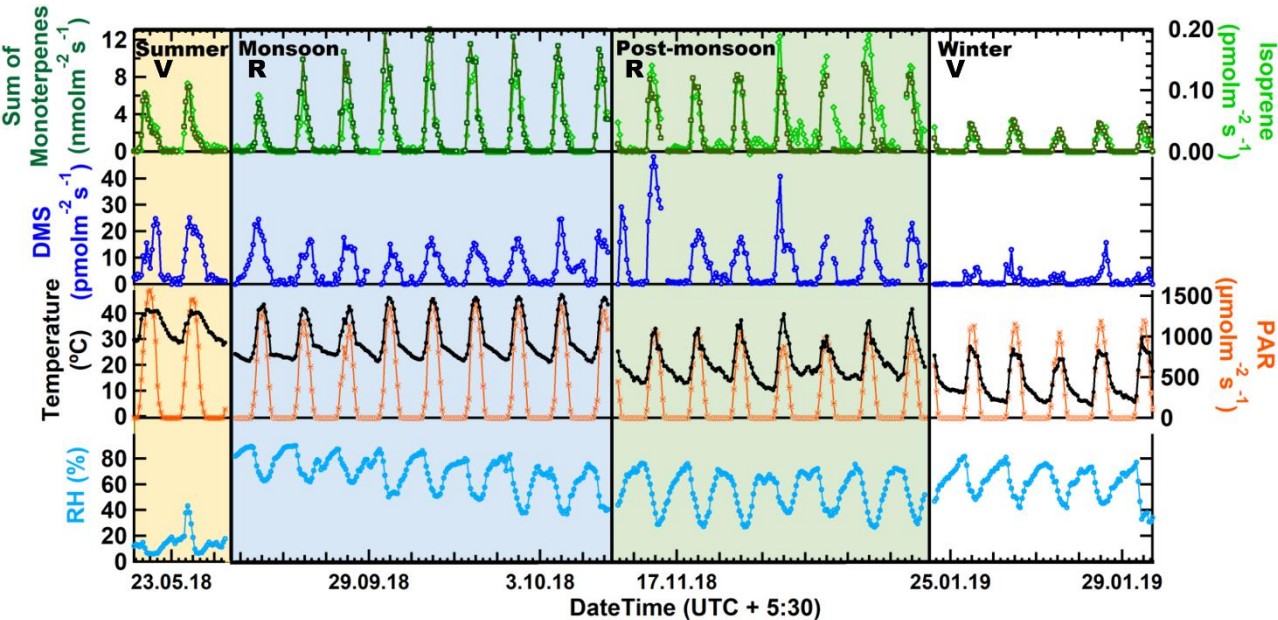

Figure 2: BVOC emission fluxes (expressed in nanomols or picomols per ($m^2$) leaf area per second) along with PAR and temperature and relative humidity. R: Reproductive growth phase V: Vegetative growth phase.

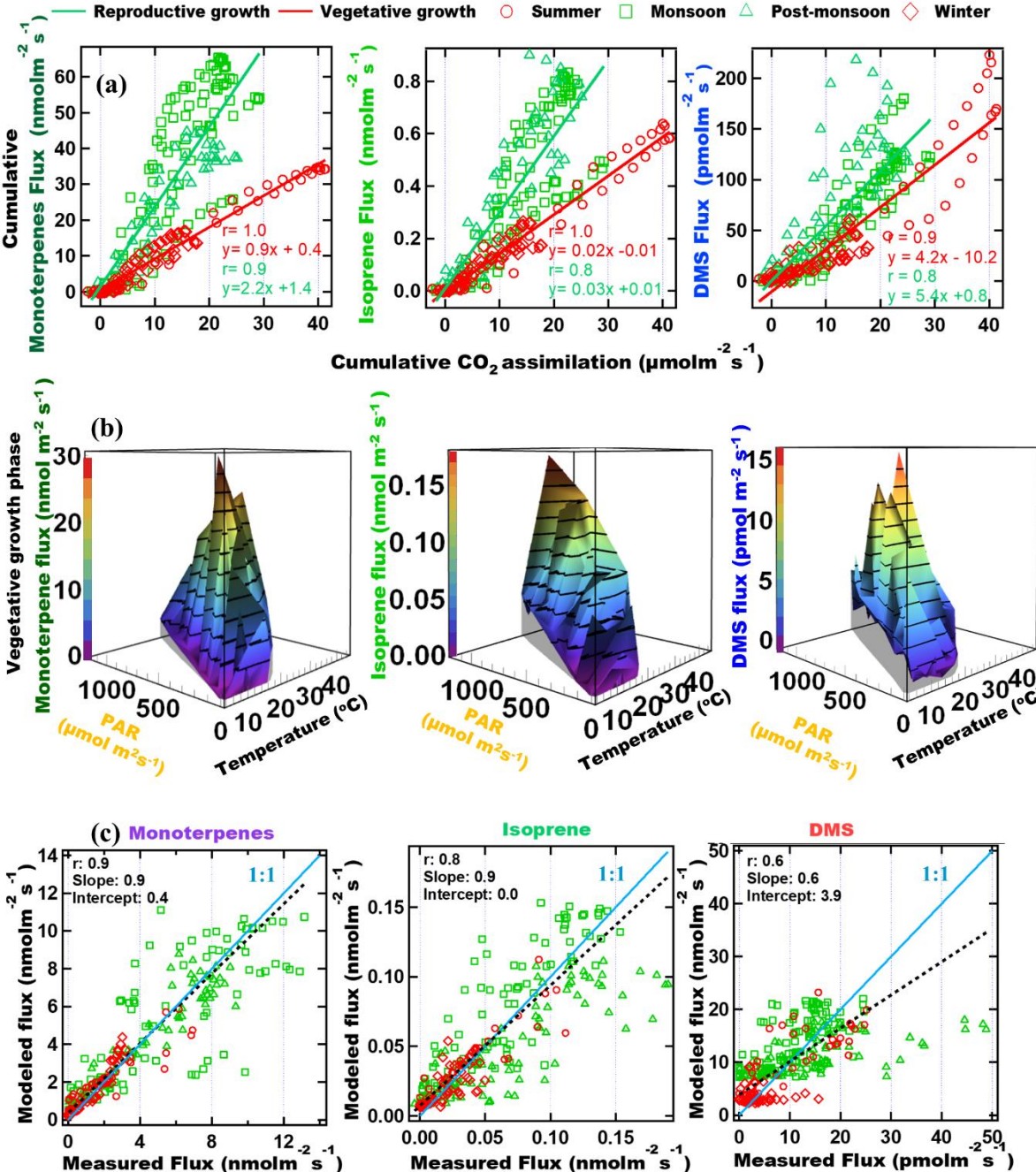

Figure 3(a)**:** Cumulative BVOC emission fluxes versus cumulative $CO_2$ assimilation. Cumulative fluxes were calculated for every hour of the day and accumulated from sunrise until that hour, (b) 3-D plot showing the correlation of the emission fluxes with instantaneous chamber temperature and PAR for vegetative growth phase and (c) Modeled versus measured VOC emission fluxes using parameterization presented in Table 3

Table 1. Summary of the sampling details of all the four trees with average temperature, photosynthetic active radiation (PAR) and variability as standard deviation of the average in parantheses

| TREE | Time period | Temperature (ºC) avg(variability) | PAR (µmol m$^{-2}$s$^{-1}$) avg (variability) |
|---|---|---|---|
| Tree 1 (Winter) | 24.01.2019-29.01.2019 | 13.5 (7.0) | 283 (408) |
| Tree 2(Winter) | 3.2.2019-4.2.2019 | 13.5 (6.1) | 252 (319) |
| Tree 3(Winter) | 5.2.2019-6.2.2019 | 19.9 (9.1) | 261 (310) |
| Tree 4 (Winter-offline) | 9.2.2019-10.2.2019 | 21.1 (12.1) | 338 (384) |
| Tree 1 (Summer) | 22.05.2018-24.05.2018 | 34.9 (4.7) | 266 (384) |
| Tree 1 (Monsoon) | 25.09.2018-04.10.2018 | 29.9 (8.0) | 232 (363) |
| Tree 1 (Post-monsoon) | 15.11.2018-22.11.2018 | 21.1 (7.1) | 170 (278) |

5 **Table 2. Average seasonal BVOCs emission fluxes from big-leaf Mahogany in different seasons normalized to the leaf dry weight alongwith variability as standard deviation of the average in parantheses.**

| Season | Monoterpene µg g$^{-1}$ hr$^{-1}$ | Isoprene µg g$^{-1}$ hr$^{-1}$ | DMS µg g$^{-1}$ hr$^{-1}$ |
|---|---|---|---|
| Summer-Avg | 6.8 (10.1) | 0.1 (0.1) | 19.2 (19.0) |
| Monsoon-Avg | 14.7 (21.6) | 0.1 (0.1) | 17.1 (17.1) |
| Post-monsoon-Avg | 7.8 (12.8) | 0.1 (0.1) | 18.8 (21.6) |
| Winter-Avg (Trees 1,2,3 4) | 2.2 (3.6) | 0.02 (0.02) | 2.9 (4.3) |

**Table 3. Bivariate fit functions and their coefficients for BVOC emission flux parameterizations as function of PAR and temperature in both the reproductive and vegetative phases of Mahogany**

Vegetative phase modeling fn:

**f(T,PAR)** = a*exp(b*PAR)+c*exp(d*T)

| | a | b | c | d |
|---|---|---|---|---|
| Monoterpenes | 0.14 | 0.003 | 0.27 | 0.10 |
| Isoprene | 0.01 | 0.002 | 0.000008 | 0.20 |
| DMS | 1.89 | 0.00001 | 0.02 | 0.16 |

Reproductive phase modeling fn:

**f(T,PAR)** = a* PAR+c*exp(d*T)

| | a | c | d |
|---|---|---|---|
| Monoterpenes | 0.009 | 0.66 | 0.01 |
| Isoprene | 0.0001 | 0.003 | 0.05 |
| DMS | 0.01 | 5.89 | 0.01 |

**Table 4. Distribution of Mahogany in natural forests and in plantations in terms of ground area, tree density, leaf area and calculated annual emission fluxes of monoterpenes, isoprene and DMS.**

| Country | Natural Area[i] ($10^4$ km$^2$) | Plantation Area[ii] (km$^2$) | Tree density[iii] Natural/Plantation ($\times 10^2$ km$^{-2}$) | Leaf area[iv] (km$^2$) | Monoterpenes (Gg yr$^{-1}$) | Isoprene (Mg yr$^{-1}$) | DMS (Mg yr$^{-1}$) |
|---|---|---|---|---|---|---|---|
| **Brazil** | 139.6 | - | 0.014-1.17[b]/- | 1564-10756 | 10-69 | 82-565 | 17-119 |
| **Peru** | 56.5 | - | - | 9042 | 58 | 475 | 100 |
| **Bolivia** | 18.9 | - | 0.1-0.2[c]/- | 1512-3025 | 9.7-19 | 79-159 | 17-33 |
| **Nicaragua** | 5 | - | 0.6/- | 2400 | 15 | 126 | 27 |
| **Mexico** | 3.6 | - | 1.0/- | 2881 | 18 | 151 | 32 |
| **Ecuador** | 3.5 | - | - | 2801 | 18 | 147 | 31 |
| **Colombia** | 2.6 | - | - | 2080 | 13 | 109 | 23 |
| **Guatemala** | 2.8 | - | 0.2-2.0/- | 448-4480 | 2.9-29 | 24-235 | 4.9-49 |
| **Honduras** | 1.7 | - | 2.0/- | 2720 | 17 | 143 | 30 |
| **Venezuela** | 1.2 | - | 1.0[d]/- | 960 | 6.1 | 50 | 11 |
| **Panama** | 1 | - | 0.1/- | 80 | 0.5 | 4.2 | 0.88 |
| **Belize** | 1 | 5.91 | 1.0-2.5/119-288[e] | 825-2061 | 5.3-13 | 43-108 | 9.1-23 |
| **Costa Rica** | 0.3 | - | 0.5-2.5/- | 120-600 | 0.77-3.8 | 6.3-32 | 1.3-6.6 |
| **Indonesia** | - | 1160 | - | 3410 | 22 | 179 | 38 |
| **Fiji** | - | 420 | - | 1235 | 7.9 | 65 | 14 |
| **Philippines** | - | 250 | - | 735 | 4.7 | 39 | 8 |
| **Sri Lanka** | - | 45 | - | 132 | 0.85 | 6.9 | 1.5 |
| **Guadeloupe** | - | 40 | - | 118 | 0.75 | 6.2 | 1.3 |
| **Martinique** | - | 15 | - | 44 | 0.28 | 2.3 | 0.49 |
| **Puerto Rico** | - | 13.81 | -/66.7-200[e] | 33-99 | 0.21-0.64 | 1.8-5.2 | 0.37-1.1 |
| **Kerala, India** | - | 1.70[a] | - | 5 | 0.03 | 0.26 | 0.06 |
| **Honduras** | - | 1.50 | - | 4 | 0.03 | 0.23 | 0.05 |
| **St. Lucia** | - | 1.00 | - | 3 | 0.02 | 0.15 | 0.03 |
| **TOTAL** | 237.7 | 1953.92 | | 33154-49674 | 212-317 | 1740-2607 | 366-548 |

[i], [ii,e]Lugo et al. (2003), [iii]Gillies et al. (1999),[a]Mohandas (2000), [b]Grogan et al. (2008), [c]Gullison et al. (1996), [d]Kammesheidt et al. (2001); Leaf Area Index: 2.94 (Jhou et al., 2017); Crown radius (m)= 0.139 x diameter (cm) - 2.82 x10$^{-4}$ x [diameter (cm)]$^2$, r$^2$ = 0.97 (Gullison et al., 1996)