# Peer review of "High DMS and monoterpene emitting big leaf Mahogany trees: discovery of a missing DMS source to the atmospheric environment"

_Atmospheric Chemistry and Physics, 2019_

## Referee Comment (RC1) · Anonymous Referee #1 · 3 Jul 2019

General comments:

This paper reports high volatile emissions from the big leaf Mahogany trees. Specifically, the authors have observed high emissions of volatiles identified as DMS and monoterpenes, and low from isoprene. By means of seasonal measurements, the authors describe the emissions as a function of environmental response such as temperature, radiation, and physiological growth phases and study the relationships with net assimilation. Using the seasonal fluxes, the authors provide the first global estimation of BVOC emissions from Mahogany trees and state that this tree-species is one of

the missing natural sources of ambient DMS over the Amazon rainforest. Overall, the subjects addressed in the manuscript are appropriate for ACP. However, I have some serious concerns concerning reproducibility and chemical identification that needs to be addressed. Please see my specific questions below.

Specific comments:

1) Number of biological replicates: The number of biological replicates (i.e. number of trees) is unclear. The authors state that they have used 4 different trees (P4L8). However, only one (n=1) was used for inter-seasonal variability (Figure 1). From the remaining 3 trees, 2 of them were measured "at high temporal resolution" online (P4L16) and 1 was sampled for "offline analysis". Overall, the biological variability of BVOC emissions is never given, and which data is presented in figure 1 and table 1-2. Please provide the variance of data dispersion of the BVOC emission based on either standard error or standard deviations (see also further comments later on).

2) Chemical identification of BVOCs: This is largely uncertain especially for isoprene and DMS. The chemical identification of VOC requires special attention, in particular for uncharacterized tree species and when measurements are based only on PTR-MS. Especially for DMS, the key compound of the actual paper, more evidence should be provided. First of all, there is little and sometimes unclear evidence of DMS emissions from trees. Second of all, a chemical identification based on PTR-QMS (Q: quadrupole), with $\sim$ 1 amu mass resolution is too little to certainly say that the volatile measured at m/z 63 is DMS. For instance, m/z 63 might originate from several VOCs, including those at MW $\sim$62g/mol or larger but fragmenting down to m/z 63. Some examples are the related dimethyl disulfide (CH3SSCH3, DMDS), and dimethyl trisulfide (CH3SSSCH3, DMTS), but possible any large compounds that fragment at m/z 63. Similarly, m/z 69 can be other than isoprene, for instance, alkyl aldehydes such as pentanal, octanal, nonanal, decanal, etc. I personally think that a large amount of m/z 137 from plants is likely to be monoterpene, so here the result is, in my opinion, reasonable because it is extremely unlikely the large emission of other unknown biogenic volatiles

at m/z 137. Although the gold technique for VOC identification remains GC-MS, VOC might be convincingly identified using PTR-MS technology by proper additional fragmentation study, or isotopic pattern simulation study, or switching the reagent ion for VOC ionization.

3) Estimation of the global annual emission of BVOCs from Mahogany: This part is very interesting but there is a fundamental contradiction. The authors state that the biomass data available (global distribution of Mahogany trees) is "by no means comprehensive" P9L27-31. So, it is unclear why the global BVOC estimations are then "meaningful".

4) Overall there is a lack of statistical analysis. When comparing the temperature and light intensities of the different season, are those significantly different? And the corresponding BVOC emissions? These and further data should be supported by appropriate statistical analysis.

5) Relative humidity: it is necessary for the reader to see the humidity data along with figure 1, essential when comparing Monsoon with post-Monsoon data.

6) Seasonality: To describe inter-seasonal variability of BVOC emissions, the authors seem to have used one single tree (n=1). This is not scientifically acceptable. It is unclear how reproducible the experiment is and what is the intra-species variance of BVOC emissions. Either the experiments are performed with more replicates (at least n=3), or the data should be removed from the manuscript.

7) Seasonality: to describe the seasonal change of BVOC emissions, the authors have performed the measurements during summer, fall (during and post-monsoon) and winter. Why did they not consider spring? The tree phenology strongly changes in spring, which is known as an important player in changing the seasonality of BVOC emissions (e.g. (Fischbach et al., 2002; Noe et al., 2012; Grote et al., 2014; Vanhatalo et al., 2018).

8) Seasonality: I find two days of measurements of one unique tree, not representative

for describing seasonal emission during summer.

9) Methods: it appears that the authors did not make any background correction using empty cuvette. To correctly calculate the BVOC fluxes, background measurements are necessary for taking into account the chemical noise of the cuvette system. However, this point does not seem to be critical since the emissions go to nearly zero during the night. However, data should be corrected before publication.

10) Table1: what are the variability of the data?

11) Table1: In the last column, what do "5" and "10" refer to?

12) Table2: why the function is different between "vegetative" and "reproductive" phase? This makes the comparison difficult. And what it the working hypothesis behind the use of these 2 functions? If available, please cite relevant literature. Overall the rationale is not described.

13) It is unclear the calibration procedure used for VOC quantification. Which molecules have been used for the calibration of the PTRMS? Was the standard mixture passing throughout the whole cuvette and canister system during calibration? If the authors have calculated the compounds-specific sensitivities, why did they sum up m/z 81+137 for monoterpene measurements?

14) M/z 81 originate also from other compounds, in particular, LOX products and sesquiterpenes. Can the authors show the correlation between 81 and 137 to rule out that other VOC were included as monoterpenes? Alternatively, the quantification should be based using only the parent ion (i.e. m/z 137).

Technical corrections:

P5L3: How long were the Teflon tubing between cuvette and sampling? Fig1: "nmol/m2 s" should be "nmol m-2 s-1" Fig2: each individual subplot should be named (e.g. (a), (b), etc. . .) Fig2: the unit of light intensity is missing Fig2: please give slopes and intercepts for the linear regressions in the first raw of subplots. Fig2 legend: please add

here the time period of the cumulative BVOC flux and assimilation. P4L6-10: please remove the number of measurements, since this information is misleading. P4L26: please describe the leafage. P5: "The input air was sampled at regular interval". It is unclear when and how often did the authors sample the inlet air. P6L29-30 Are these temperatures statistically different? P6L30-31: Are the light intensities statistically different? P8L31: "paramterization" should be "parameterization"

Fischbach RJ, Staudt M, Zimmer I, Rambal S, Schnitzler J-P. 2002. Seasonal pattern of monoterpene synthase activities in leaves of the evergreen tree Quercus ilex. Physiologia plantarum 114: 354–360.

Grote R, Morfopoulos C, Niinemets Ü, Sun Z, Keenan TF, Pacifico F, Butler T. 2014. A fully integrated isoprenoid emissions model coupling emissions to photosynthetic characteristics. Plant, Cell and Environment 37: 1965–1980.

Noe SM, Hüve K, Niinemets Ü, Copolovici L. 2012. Seasonal variation in vertical volatile compounds air concentrations within a remote hemiboreal mixed forest. Atmospheric Chemistry and Physics 12: 3909–3926.

Vanhatalo A, Ghirardo A, Juurola E, Schnitzler JP, Zimmer I, Hellén H, Hakola H, Bäck J. 2018. Long-term dynamics of monoterpene synthase activities, monoterpene storage pools and emissions in boreal Scots pine. Biogeosciences 15: 5047–5060.

---

## Referee Comment (RC2) · Anonymous Referee #2 · 11 Jul 2019

Review of High DMS and monoterpene emitting big leaf Mahogany trees: discovery of a missing DMS source to the atmospheric environment

This manuscript presents 24 (+6 in SI) days of measurements of monoterpene, isoprene and DMS emissions from Mahogany in India. The measurements were conducted using a PTR-Quad and a dynamic branch chamber. The results were compared with modelled emissions and then globally upscaled. The measurements identify Mahogany as one of the missing sources of DMS in the rainforest.

The presented data is novel and even though isoprene and monoterpene emission measurements are more and more common, DMS emission measurements are rare. Therefore I see the result as interesting for the scientific community and recommend the paper to be accepted.

However, I would like that the Authors addressed following comments before publication:

- One major flaw in the manuscript is the missing discussion about the challenges when measuring DMS (and isoprene). Even though the authors have cited literature (de Gouw et al., 2006; Jardine et al., 2015) which extensively discuss the problem of acetaldehyde clusters' possible influence on mass 63, there is no evidence in this manuscript that this influence was ruled out. Even though, Jardine et al. (2015) stated that in their setup no influence could be seen, their instrumental settings seemed to have been optimized to suppress waterclusters ($\frac{H_2O \cdot H_3O^+}{H_3O^+}$ < 4%; E/N>145 -> please see my comment P6 L13).
  There is a sentence (P6 L10-16) stating that isoprene and DMS can be measured at their respective masses without much fragmentation, which is true. As long as the PTR-MS is frequently calibrated under measurement conditions, fragmentation losses (of e.g. isoprene or DMS) are corrected by the sensitivity (this is just the case if the measured and calibrated compound are identical and the signal is above the limit of detection). However, other compounds fragmenting/clustering on the same mass (e.g. M69, M63) are a major source of uncertainty. And therefore identifying/ruling out a possible influence of acetaldehyde to the DMS signal (M63) is crucial. There is a similar issue with MBO, which fragments to M69.
- It is not very clear when the offline sampling was used. The only references to the offline sampling are in the methods part and in the SI. I concluded, that all data used in the main manuscript were online. Therefore, I would recommend, moving the description of the offline sampling to the SI.
- P2 L6-8 (also P3 L20-22): Please rephrase (South, Central and North America -> Americas; atmospheric environments -> environments).
- Refer from using the word 'fluxes' (= bidirectional) when you discuss your measurements. As your setup cannot capture deposition, use the word 'emissions' instead (e.g. P4 L2, P4 L5, P6 L28, P6 L31,...)
- P4 L6-8: The number of measurements sounds impressive, however it is not clear what those measurements are. Is a measurement the measured 1 s dwell time data point? Is it one cycle through all measured compounds? ... If the authors want to state the amount of data at all, I would suggest to state the number of 1 h data points, shown e.g. in Fig. 1.
- P4 L9: Omit outdoor (there is no natural indoor environment for Mahogany, I assume).

- P4 L29: … *using a series of traps containing steel wool, silica gel and activated charcoal*. If those traps were custom build, please state so, otherwise please add the type and brand (this information can be very helpful for people who want to use a similar setup).
- P5 L2: … *using a second pump by ensuring to have a small positive pressure inside the chamber*…How was this positive pressure achieved (by regulating the flow with a MFC or does this pump have a flow lower than 30 L min$^{-1}$) and how large was the flow flushing the 60-65 m inlet line?
- P5 L8: *This is significantly longer than the steady-state* … this statement is correct, however, after installing the chamber a longer equilibration time is necessary to prevent measurement artefacts of physical stress or small injuries of the branch (caused by the installation of the cuvette).
- P6 L6: …*dwell time of 1 s at each m/z channel*. Which channels were measured (stated are M63, M69, M81, M137, however I assume also instrumental background peaks (e.g. M21, M25, M32, M45, M39, M87) were measured for quality assessment) and what was the measurement cycle time (or what was the sampling frequency)?
- P6 L8: Please state the compounds in the calibration gas, as well as the uncertainty of the calibration gas. Furthermore please state average sensitivities with standard deviation and limit of detection (concentration and emission) for the main compounds (Isoprene, DMS and the calibrated monoterpene compound).
- P6 L9: *The total measurement uncertainty was less than 10% for isoprene and DMS and less than 15% for the sum of monoterpenes* ...This low uncertainty seems for me very optimistic for the used setup, especially after addressing my first comment. If I remember correctly the calibration gas from Appel-Riemer has an uncertainty of 5% (valid for 1 year after filling of the gas bottle), then adding uncertainties for 3 MFCs (1 for the inlet at the chamber, 2 for calibration, I assume), 60 – 65 m of tubing, an extra pump, as well as only 1 calibration per season (up to 20 days before measurements). Could the authors please provide the uncertainty calculations for those numbers? Also, are the same values valid for the offline sampling?
- P6 L13: I could not find any statement about the used E/N in Jardine et al. (2015). However, from the stated values in their paper ($p_{drift}$= 2.0 mbar, $V_{drift}$= 600 V) it seems to be either 145 Td (if the drift temperature was 50°C, like their heated inlet) or 149 Td (if the drift temperature was the more common 60°C). Therefore, their measurements did not fall under the standard operational conditions between 130 and 135 Td. Could the authors please provide average and maximum $\frac{H_2O \cdot H_3O^+}{H_3O^+}$ ratios during this measurement campaign (i.e. was it comparable with the 4% in Jardine et al., 2015)?
- P6 L17: Also state the used drift voltage.
- P7 L3-4: Please add that this statement is valid for winter, as below it is stated that *photosynthetically fixed carbon* (normally associated with PAR) *may be more important than emissions from storage pools* (normally associated with T).
- P7 L20: Remove the space in the hyperlink (…ac.uk\cnhgroup…)
- P7 L29: Add the year to the Jardine et al. citation.
- P8 L24-25: *It is also evident that DMS has a strong dependence on temperature, but not on PAR*. The PAR dependency is very hard to see in Fig 2(b). It seems to me quite difficult to state anything about DMS, as there is a rather high offset at low PAR and T (it seems that temperature has no effect at low PAR), there is a huge decline at high T when PAR is around 500 µmol m$^{-2}$ s$^{-1}$.
- P10 L12: Omit outdoor (see earlier comment).

- P14 Figure 1: It seems the PAR axis label has a different color that the PAR graph, please use the same color.
- P15 Figure 2:
  - (a) Please provide the slopes for the linear fits
  - (b) Sadly, these plots are not very clear. They give an idea of the temperature dependence, however, the PAR dependence cannot be seen. I recommend either turning these 3D plots to have the origin (0-point) at the bottom middle and PAR and T axis going with the same angles left and right (=symmetrical) or changing the plot style altogether. Depending on the changes, please state in the figure caption that the color corresponds to the respective VOC flux. Also to make that more obvious, the color bars could be stretched to cover the whole axis (then maybe one set of axis labels would be enough).
  - (c) Here I would recommend changing all y axes labels to Modeled flux (with respective unit) and state the compound as a title (centered above each plot)
- P16 Table 2: I recommend renaming the variables in the *Reproductive phase modeling fn:* **f(T,PAR)**=a*PAR+c*exp(d*T) to make it easier to compare to the vegetative phase modeling fn.
- P17 Table 3: *(x100 nos./km$^2$)*; Please use 100 (or 10$^2$) if x100 is a multiplication, similar as in the second column in this table. Please clarify '*nos.*' Is it a unit? If so, please explain it in the table caption.
- PS2 Figure S1: The offline canister sampling is twice in your schematic (once at the KNF pump2 and once behind it). Normally the tubing is marked by 2 lines (e.g. between MFC and Tedlar bag), however right before the instruments it changes to normal arrows, please use only one style.
- PS3 & 5, Figure S3 & S5: The figure caption seems to be wrong, as it says that BVOC emissions are shown, but the y axis unit is ppb. Assumingly it should state Time series of BVOC concentrations in the chamber with corresponding… (in case there are actually emissions shown, please calculate background emissions and change the label).
- PS4 & 5, Figure S4 & S5: As there is no reference in the main manuscript to these figures, could you please provide some context to the figures.

Technical:

(see: https://www.atmospheric-chemistry-and-physics.net/for_authors/manuscript_preparation.html)

Use SI units (e.g. P4 L19: change to metric)
[*For units of **physical quantities**, the metric system is mandatory and, wherever possible, SI units should be used.*]

State the inner diameters of tubing (instead of outer diameters), as those are the crucial parameters for volume, residence time, line losses.

Follow the recommendations of the journal for the format of your units (e.g. P2 L13&14, P5 L1,…)
[*Regarding the **notation**, if units of physical quantities are in the denominator, contain numbers, and are abbreviated, they must be formatted with negative exponents (e.g. 10 km h$^{-1}$ instead of 10 km/h)*]

Unify the way you state instruments (often type, company, country are stated, sometimes not; e.g. P4 L13, P4 L25, P5 L1, P5 L5,…)

Please use either L or l for the unit of liter (e.g. P5 L1, P5 L15,…)

Sometimes spaces are missing between values and units (e.g. P4 L15,P9 L21,… )

---

## Author Comment (AC1) · 12 Sep 2019

**High DMS and monoterpene emitting big leaf Mahogany trees: discovery of a missing DMS source to the atmospheric environment**

We greatly appreciate the positive remarks and constructive suggestions by the reviewers for improving the manuscript. In accordance with the valuable suggestions, we have undertaken the revisions. We have uploaded the revised manuscript in its final version along with a version that has the "track changes" mode for convenience and easy perusal to this response file.
We are confident that the revised manuscript has addressed all the valid concerns raised by the reviewers and hope that the revised submission may now be considered suitable for publication in ACP.
Best Regards,
Vinayak Sinha (on behalf of all authors)

Please find the revisions/replies (in blue) to the specific points (**in black**) raised by the esteemed reviewers for easy perusal.

**Anonymous Referee #1**

**General comments**: This paper reports high volatile emissions from the big leaf Mahogany trees. Specifically, the authors have observed high emissions of volatiles identified as DMS and monoterpenes, and low from isoprene. By means of seasonal measurements, the authors describe the emissions as a function of environmental response such as temperature, radiation, and physiological growth phases and study the relationships with net assimilation. Using the seasonal fluxes, the authors provide the first global estimation of BVOC emissions from Mahogany trees and state that this tree-species is one of the missing natural sources of ambient DMS over the Amazon rainforest. Overall, the subjects addressed in the manuscript are appropriate for ACP. However, I have some serious concerns concerning reproducibility and chemical identification that needs to be addressed. Please see my specific questions below.

**Reply:** We are grateful to the reviewer for her/his careful reading of the manuscript and comments deeming that the manuscript is appropriate for ACP subject to addressing the major and minor comments raised in the esteemed and insightful review.

**Specific comments:**
**1)** Number of biological replicates: The number of biological replicates (i.e. number of trees) is unclear. The authors state that they have used 4 different trees (P4L8). However, only one (n=1) was used for inter-seasonal variability (Figure 1). From the remaining 3 trees, 2 of them were measured "at high temporal resolution" online (P4L16) and 1 was sampled for "offline analysis". Overall, the biological variability of BVOC emissions is never given, and which data is presented in figure 1 and table 1-2. Please provide the variance of data dispersion of the BVOC emission based on either standard error or standard deviations (see also further comments later on).

**Reply:** We regret that these details were not clear in our original submission due to the information being scattered in different places (e.g. figures in main paper and supplement).
In order to make the above clear in the revised version and also considering the esteemed reviewer's other points we have made the following changes in the revised version by adding a new Table (new Table 1 of revised MS and also shown below for easy perusal):

**Table 1.** Summary of the sampling details of all the four trees with average temperature, photosynthetic active radiation (PAR) and variability as standard deviation of the average in parantheses.

| TREE | Time period | Temperature (ºC) avg (variability) | PAR ($\mu$mol m$^{-2}$ s$^{-1}$) avg (variability) |
|---|---|---|---|
| Tree 1 (Winter) | 24.01.2019-29.01.2019 | 13.5 (7.0) | 283 (408) |
| Tree 2 (Winter) | 3.2.2019-4.2.2019 | 13.5 (6.1) | 252 (319) |
| Tree 3 (Winter) | 5.2.2019-6.2.2019 | 19.9 (9.1) | 261 (310) |
| Tree 4 (Winter-offline) | 9.2.2019-10.2.2019 | 21.1 (12.1) | 338 (384) |
| Tree 1 (Summer) | 22.05.2018-24.05.2018 | 34.9 (4.7) | 266 (384) |
| Tree 1 (Monsoon) | 25.09.2018-04.10.2018 | 29.9 (8.0) | 232 (363) |
| Tree 1 (Post-monsoon) | 15.11.2018-22.11.2018 | 21.1 (7.1) | 170 (278) |

Thus, during the winter season, four replicates were sampled as shown in the Table above.

The above information has now been added to the revised MS in Section 2.1, Paragraph 1; Lines 1-18 as follows:

*"Table 1 provides a summary of the sampling dates alongwith the average and ambient variability (as standard deviation) of the temperatures and photosynthetic active radiation (PAR) during each of the sampling experiments."*

And by re-writing lines 6-9 at Page 4 (Section 2.1) of the original submission as follows:

*"While sampling and biogenic VOC emission measurements were performed from four Mahogany trees in winter (details in Table 1), the sampling and biogenic VOC emission measurements for three other seasons were from one of the four trees (namely Tree 1 in Table 1) as follows: 2018 summer from 22-24 May (n=52 hours of measurements), 2018 monsoon (n=200 hours of measurements) from 25 September-4 October, 2018 post-monsoon (n=163 hours of measurements) from 15-22 November, and 2019 winter from 24-29 January (n=120 hours of measurements)."*

As suggested by the reviewer we have also added additional information on the average hourly wintertime fluxes and variability (as standard deviation) for trees 2, 3 and 4 alongwith the average flux and variability observed for tree 1 (Figure 1 in revised MS).

This new Figure is also shown below for easy perusal:

[Figure]

**Figure 1 of response and new Figure 1 of the revised submission.** Average wintertime fluxes and variability (as standard deviation) for trees 2, 3 and 4 shown in comparison to average flux and variability of tree 1

The new discussion pertaining to the above Figure has now been added at the start of the Results and Discussion as follows:

*"Figure 1 shows the average wintertime fluxes and variability (as standard deviation) from trees 2, 3 and 4 shown in comparison to the average flux and variability of tree 1. Earlier in Table 1, a summary of the sampling, PAR and temperature data for these experiments has already been provided. It can be observed  that the observed hourly fluxes from Tree 1 (which was also sampled in three other seasons as mentioned in Section 2.1) were always within the observed one sigma variability of the fluxes for monoterpenes and isoprene obtained  from Trees 2, 3 and 4 the observed hourly fluxes from Tree 1 (which was also sampled in other seasons) for monoterpenes and isoprene were always within the observed one sigma variability of the fluxes determined from Trees 2, 3 and 4.  For DMS, the observed daytime fluxes from Tree 1 were at times lower than the 1 sigma variability range of the DMS flux observed from Trees 2, 3 and 4, and at the lower end of the observed fluxes from the other trees. This implies that the DMS fluxes obtained using Tree 1 do not overestimate the DMS fluxes for Swietenia macrophylla. Overall, based on comparison with three other replicate trees of Mahogany*

*(trees 2, 3 and 4) for the wintertime data, one can surmise that there is no evidence of Tree1's emission profile and fluxes being anomalous."*

In appreciation of the reviewer's further suggestion, in the revised Table 1 of original submission (which is now Table 2 of the revised submission), we now also include the variability (as 1 sigma standard deviation) for the seasonally averaged fluxes, and report the average flux from all four trees (n=4) for winter.

The revised Table 1 of the original submission (now new Table 2 of revised submission) is shown below for easy perusal:

**Table 2.** Average seasonal BVOCs fluxes from big-leaf Mahogany in different seasons normalized to the leaf dry weight alongwith variability as standard deviation of the average in parantheses.

| Season | Monoterpene µg/g/hr | Isoprene µg/g/hr | DMS ng/g/hr |
|---|---|---|---|
| Summer-Avg | 6.8 (10.1) | 0.1 (0.1) | 19.2 (19.0) |
| Monsoon-Avg | 14.7(21.6) | 0.1(0.1) | 17.1 (17.1) |
| Post-monsoon-Avg | 7.8 (12.8) | 0.1 (0.1) | 18.8 (21.6) |
| Winter-Avg (Trees 1,2,3 4) | 2.2 (3.6) | 0.02 (0.02) | 2.9 (4.3) |

In response to the reviewer's comment about replicates, when the authors were preparing this response, they carefully also looked for the number of replicates studied in one of the previous pioneering DMS emission study by Jardine et al. 2015 who reported DMS emission data from 7 tropical trees. We could not find mention of the number of replicates that were sampled, and it appears that in those experiments also there was just one tree like in our study, the data in some instances was also only 4 days and no seasonal emission studies were reported for the DMS emission from those 7 trees. In spite of these constraints/limitations, the data from that study has been extremely helpful for us to interpret and discuss the present results. While we do appreciate the reviewer's point that sampling more replicates in other seasons would have been better and will make due note of this in the conclusions, the data from the four replicates (available for the winter season) does demonstrate that Tree 1 was certainly not anomalous with regard to BVOC emissions of monoterpenes, isoprene and DMS. There is therefore great value in reporting the seasonal data, considering the paucity of what is known about DMS seasonal emissions from trees (this study to the best of our knowledge contains first such information on seasonal emission tendencies). We are grateful that the reviewer's points have helped revise the original submission to clarify and strengthen these and related points.

We have added a new paragraph to the Conclusions section on Page 10 of original submission as follows:
*"While several novel insights have been obtained from this study such as discovery of a new terrestrial source with high emissions for monoterpenes and DMS relative to other known terrestrial sources, one limitation has been the lack of data from replicates for three of the four seasons. Based on comparison with three other replicate trees of Mahogany (Trees 2, 3 and 4) for the wintertime data, one can surmise that there is no evidence of Tree1's emission profile and fluxes being anomalous and hence considering the paucity of what is known about DMS*

*seasonal emissions from trees (this study to the best of our knowledge contains first such information on seasonal emission tendencies), the insights about seasonality of Mahogany emissions obtained in this study are also valuable. We acknowledge, however that data from more replicates would be better to characterize the intra-species and seasonal emissions variability better and should be addressed in future studies."*

**2)** Chemical identification of BVOCs: This is largely uncertain especially for isoprene and DMS. The chemical identification of VOC requires special attention, in particular for uncharacterized tree species and when measurements are based only on PTR-MS. Especially for DMS, the key compound of the actual paper, more evidence should be provided. First of all, there is little and sometimes unclear evidence of DMS emissions from trees. Second of all, a chemical identification based on PTR-QMS (Q: quadrupole), with~1 amu mass resolution is too little to certainly say that the volatile measured at m/z 63 is DMS. For instance, m/z 63 might originate from several VOCs, including those at MW~62g/mol or larger but fragmenting down to m/z 63. Some examples are the related dimethyl disulfide (CH3SSCH3, DMDS), and dimethyl trisulfide (CH3SSSCH3, DMTS), but possible any large compounds that fragment at m/z 63. Similarly, m/z 69 can be other than isoprene, for instance, alkyl aldehydes such as pentanal, octanal, nonanal, decanal, etc. I personally think that a large amount of m/z 137 from plants is likely to be monoterpene, so here the result is, in my opinion, reasonable because it is extremely unlikely the large emission of other unknown biogenic volatiles at m/z 137. Although the gold technique for VOC identification remains GC-MS, VOC might be convincingly identified using PTR-MS technology by proper additional fragmentation study, or isotopic pattern simulation study, or switching the reagent ion for VOC ionization.

**Reply:** We thank the reviewer for his/her critical comment about the chemical characterization of BVOCs especially since this tree was an uncharacterized tree species. We regret that we missed adding sufficient detail and discussion regarding the steps we took to address these known issues concerning identification of DMS and isoprene using a PTR-QMS in the original submission, as we wanted to keep the technical description short.
We regret the confusion to both reviewers 1 and 2 and readers due to it. Below please find the detailed discussion of how we conducted our experiments which addresses these technical issues /potential interferences raised both by reviewer 1 and 2:
When we commenced the first set of plant cuvette measurements in summer we undertook mass scans for the input air and output air into the branch cuvette over the entire mass range of (m/z 21- m/z 210) during the experiments. We found that in comparison to the ambient air, the mass scans contained very few peaks and the spectra was remarkably simple. A typical 30 min averaged mass scan of the output air from the branch cuvette system during the afternoon period (which was characterized by highest signals and concentrations) after subtracting the input air mass signal at the m/z channel is shown below.

**Figure 2 of response and new Figure S3**: A typical 30 min averaged PTR-MS mass scan of the output air from the branch cuvette system during the afternoon period after subtraction of input background air signals showing the ion signals observed in the mass range m/z 40 to m/z 210.

[Figure]

It can also be seen that the observed isotopic of abundances of the shoulder peaks of m/z 137 ion due to the natural C-13 abundance which should be 1.1% for each C-12 atom in the ion, and therefore 11.1% of m/z137 ion abundance at m/z138 and 6.6% of the m/z81 ion abundance, respectively are entirely consistent with the ion peaks being due to monoterpenes.

The results of these mass scans formed the basis for our choice of what masses to monitor in subsequent plant chamber experiments in other seasons from the same tree. Despite the simple spectra obtained in our mass scan results during summer, for subsequent experiments conducted in the selected ion monitoring (SIM) mode in other seasons, we still monitored 60 m/z channels of interest keeping in mind the PTR-MS literature for BVOC emissions, major atmospheric VOCs, and abundant ions formed generally due to the ion chemistry in the PTR-MS drift tube, which include impurity ions such as m/z30 ($NO^+$), m/z32 ($O_2^+$) etc.. . The list of 60 also included m/z42, m/z43, m/z44, m/z45, m/z46, m/z47, m/z48, m/z49, m/z55, m/z57, m/z58, m/z59, m/z60, m/z61, m/z63, m/z65, m/z67, m/z68, m/z69, m/z70, m/z71, m/z72, m/z73, m/z74, m/z75, m/z79, m/z81, m/z83, m/z85, m/z87, m/z88, m/z89, m/z91, m/z93, m/z95, m/z97, m/z99, m/z100, m/z101, m/z105, m/z107, m/z109, m/z119, m/z121, m/z123, m/z129, m/z135, m/z137, m/z149, and m/z205. This enabled us to examine also scope for any potential new interferences due to fragmentation/clustering effects or even new emissions.

**DMS identification: addressing known issues and reviewer's concerns**

We undertook correlation of all other monitored m/z at which measurable signal was observed with the m/z63, but found no significant correlation ($r^2 \leq 0.2$) with any of them, which suggested that fragmentation of a larger volatile detected at higher mass to charge ratio was likely not responsible for the observed m/z63 signal. Reviewer 1 raised an interesting concern about the potential for dimethyl disulfide ($CH_3SSCH_3$, DMDS), and dimethyl trisulfide ($CH_3SSSCH_3$, DMTS), fragmenting and contributing to the m/z 63 signal. We note that the parent ions of these compounds would be m/z95 and m/z127. As mentioned above we did monitor m/z 95 in all the seasons but didn't monitor m/z 127 in the experiments after summer season as we didn't see any signal at this m/z in the output air of the cuvette. We also could not find any previous report suggesting the possibility of these compounds fragmenting to m/z63 under standard operating conditions of the PTR-MS such as 135 Td at which we operated our PTR-QMS. On the contrary, a recent relevant study conducted using both GC-MS and PTR-TOF-MS (under similar range of operating conditions; 120-140 Td) for organosulfur compounds which included these compounds (Perraud et al., 2016), showed that dimethyl disulfide ($CH_3SSCH_3$, DMDS), and dimethyl trisulfide ($CH_3SSSCH_3$, DMTS) DO NOT fragment and contribute to the m/z63 channel. Our own mass scans and analyses of correlated

masses were also consistent with these findings and so we were able to rule out the possibility of such higher compounds fragmenting and contributing to the m/z63 signal in our dataset.

The issue of hydration of protonated acetaldehyde which can form the following ion: $H^+$ ($CH_3CHO$) $H_2O$ (which has m/z 63) and therefore can contribute to the m/z 63 attributed to DMS indeed required careful attention as was acknowledged in our original submission too. This issue was first pointed out in the review by de Gouw and Warneke 2007 and further addressed adequately in the work by Jardine et al. 2015. The interference from this ion can be significant when both the hydrated hydronium ion and acetaldehyde concentrations are high leading to appreciable formation of $H^+$ ($CH_3CHO$) $H_2O$ in the drift tube from reactions of the $H^+$ ($CH_3CHO$) with ($H_3O^+H_2O$) ion. As pointed out by reviewer2 and by the work of Jardine et al. 2015, if the abundance of the hydrated hydronium ion ($H_3O^+H_2O$) is therefore kept to just a few percent of the primary reagent ion namely $H_3O^+$ ion (circa 4 %), which can be accomplished by operating at high Townsend ratios (e.g. $\geq$ 135 Td; 600 V and 2.2 mbar drift tube pressure), then at mixing ratios of less than 19 ppb acetaldehyde that occur in most ambient environments and well ventilated cuvette systems, this interference has been shown to be negligible (see for example results reported in the paper by Jardine et al. 2015, where even at acetaldehyde mixing ratios as high as 19 ppb, there was no measurable change in the m63 ion signal). We therefore took the above precaution of operating under high Townsend ratios in the drift tube to minimize conditions that favour formation of clusters ions by enhancing kinetic energy of the reagent ions. During all our experiments, acetaldehyde mixing ratios were below 12 ppb and under our operating conditions (135 Td), the average $H_2O \cdot H_3O^+$ to $H_3O^+$ ratio was only 4.12 % for the entire dataset which is comparable to the 4% or lower abundance during experiments conducted by Jardine et al., 2015. Our dataset was also carefully examined for indications of this potential interference biasing our measured m63 dataset. For this we plotted the 4 min averaged temporal resolution primary data for m63 ion (presented in our original submission in Fig S2) against the corresponding co-measured 4 min averaged temporal resolution primary m/z45 ion data for all the seasons.

Reference
Perraud, V., Meinardi, S., Blake, D. R., and Finlayson-Pitts, B. J.: Challenges associated with the sampling and analysis of organosulfur compounds in air using real-time PTR-ToF-MS and offline GC-FID, 10.5194/amt-9-1325-2016, 2016.

The results are shown in the Figure below.

**Figure 3 of response and new Figure S4**: **Figure S4**: Correlation of m/z63 ion signal with m/z 45 ion signal for all seasons

[Figure]

We found no significant correlation between the two (r = 0.22) and even at high m45 mixing ratios of 10 ppb, low m63 mixing ratios of 0.2 ppb occurred frequently, which would not have been the case if the m63 originated primarily from the acetaldehyde hydrated water ion cluster. Therefore, in view of the above, just like Jardine et al. 2015, we are confident that the potential interference of acetaldehyde on the DMS data was absent/negligible.

**Isoprene identification: addressing known issues and reviewer's concerns**

We fully agree that attribution of isoprene to m69 in the PTR-MS requires careful attention and consideration of known interferences from isobaric compounds and fragments of higher ions. As mentioned in the excellent review by Yuan et al. 2017 (which we also referred to and cited in the original submission) and pointed out by the esteemed reviewers, several compounds can present substantial interferences in various environments, such as furan in biomass-burning plumes, cycloalkanes in urban environments and oil/gas regions, 2-methyl-3-buten-2-ol (MBO) in pine forests, and methylbutanals and 1-peten-3-nol from leaf-wound compounds. We examine one by one each of these possible interferences for the isoprene measurements reported in our dataset. Firstly, we note that many of the potential interferences that can affect

the m69 signal while sampling ambient air influenced by mixed combustion and biogenic sources are not relevant for our experimental set up as the output air from the branch cuvettes (after subtracting input air) is exclusively influenced by only biogenic emissions. Concerning the other biogenic emissions that could still be responsible for contributing to the m69 signal measured by the PTR-MS, actually we did carry out isoprene measurements using a TD-GC-FID system simultaneously. We did not report this TD-GC-FID data for isoprene in the original submission as during tests, variable transfer losses were suspected to have occurred in the system likely within the pre-concentration unit or during transfer from the trap to the column within the TD-GC-FID system. This prevented accurate quantification of isoprene and unfortunately, it has not been possible to correct for these effects reliably. Nonetheless, even the semi-quantitative data is useful and sufficient to show that the air from the branch cuvette contained isoprene. Below we show the chromatographic peak for isoprene from the output air of the branch cuvette system identified based on the retention time of isoprene vapours that were sampled under identical operating conditions with the TD-GC-FID. The isoprene data co-measured with the TD-GC-FID for the monsoon season along with the isoprene data measured with the PTR-QMS, was found to have excellent correlation of the PTR-MS isoprene signal but the absolute values were much lower due to the suspected losses within the TD-GC-FID system. When combined with the observed diurnal variability of the m69 signal with PAR and temperature, and these additional observations using the TD-GC-FID, therefore we could not identify any known compound other than isoprene that could satisfy all the above criteria. Hence m/z 69 was confidently assigned to isoprene.

Figure 4 of response and new Figure S5: Sample chromatogram of the isoprene peak resolved on the Alumina PLOT column at a retention time of 37.5min in the air collected from the plant chamber experiment overlayed with the peak from pure isoprene vapours injected to determine the retention time of isoprene.

[Figure]

Figure 5 of response and new Figure S6. Times series of hourly averaged isoprene measurements from PTR-MS and TD-GC-FID for monsoon season

[Figure]

Figure 6 of response and new Figure S7. Correlation of hourly averaged isoprene measurements from PTR-MS and TD-GC-FID for monsoon season

[Figure]

**We have now added all this information to revised manuscript and the supplement as follows:**

*Added to revised manuscript are the following new paragraphs to Section 2.2 entitled 
[revised manuscript text omitted]

*Isoprene was detected in output air from the branch cuvette using a gas chromatograph equipped with a flame ionization detector (GC-FID 7890B, Agilent Technologies, Santa Clara, United States) which is coupled to a thermal desorption unit (CIA Advantage-HL and Unity 2, Markes International, UK) for sampling and pre-concentration. Water in the sample air was removed using a nafion dryer which also removed the oxygenated VOCs such as alcohols, aldehydes and ketones (Badol et al., 2004; Gros et al., 2011). 1000ml of dry sample air was pre-concentrated at -30°C at 20 mL/min on an ozone precursor trap (U-T17O3P-2S, Markes Internatioal, UK) which was then thermally desorbed by rapid heating to 325°C. The desorbed analytes were then transferred onto the GC instrument via a heated inlet (130°C) line. The GC instrument consisted of a capillary column (Alumina Plot, Al2O3/Na2SO4, 50m x 0.32mm, 8 μm film thickness, Agilent Technologies, Santa Clara, United States). The oven temperature was ramped from 30°C (hold for 12 min) to 200°C at the rates of 5°C/min (upto 170°C) and 15°C/min (upto 200°C) for resolving the peaks.*

*Isoprene was resolved on Alumina PLOT column at a retention time of 37.5min and identified based on the retention time of isoprene vapours injected into the TD-GC-FID system under identical instrument operational conditions as the sample. The eluted isoprene was then detected using the FID. Unfortunately due to the suspected transfer losses within the GC system, which could not be corrected, the data is only semi-quantitative and hence reported in arbitrary units.*

References:

Badol, C., Borbon, A., Locoge, N., Léonardis, T., and Galloo, J.-C.: An automated monitoring system for VOC ozone precursors in ambient air: development, implementation and data analysis, Analytical and bioanalytical chemistry, 378, 1815-1827, https://doi.org/10.1007/s00216-003-2474-0, 2004.

Gros, V., Gaimoz, C., Herrmann, F., Custer, T., Williams, J., Bonsang, B., Sauvage, S., Locoge, N., d'Argouges, O., and Sarda-Estève, R.: Volatile organic compounds sources in Paris in spring 2007. Part I: qualitative analysis, Environmental Chemistry, 8, 74-90, https://doi.org/10.1071/EN10068, 2011.

*New Figure S8: Correlation between observed m81 and m137 signals from the plant chamber output air for all seasons*

[Figure]

**3)** Estimation of the global annual emission of BVOCs from Mahogany: This part is very interesting but there is a fundamental contradiction. The authors state that the biomass data available (global distribution of Mahogany trees) is "by no means comprehensive" P9L27-31. So, it is unclear why the global BVOC estimations are then "meaningful".

**Reply:** Thank you for appreciating our effort in the estimation of global annual emission of BVOCs from Mahogany. We apologise for the confusion caused by ill choice of words. We would like to clarify that as presented in Table 3 of the original submission, which lists the peer reviewed literature used for assessing the biomass, the global BVOC estimates are comprehensive. Overall the discussion and analysis concerning the global annual emissions are quite meaningful for assessing which areas/regions of the world this hitherto unreported source may have significant impacts on for atmospheric chemistry and air quality through biogenic emissions, and therefore provide guidance for planning future BVOC emissions field studies in some of these understudied regions such as South America and Indonesia.
*Thanks to the reviewer's comment however we now realize that "by no means comprehensive" is misleading and we have rephrased P9L27-31 which now reads as follows:*

*"We would like to point out that this estimate is based on the current available information but there may be some underestimation as there are some areas where cultivation of Mahogany trees is known to occur (e.g. Jim Corbett national park in India), for which, however, accurate Mahogany biomass estimates are not available and which hence were not included in Table 3."*

**4)** Overall there is a lack of statistical analysis. When comparing the temperature and light intensities of the different season, are those significantly different? And the corresponding BVOC emissions? These and further data should be supported by appropriate statistical analysis.

**Reply:** Thank you for this valuable feedback. We now add the results of the statistical analyses as follows: We performed the Kruskal-Wallis test since it is a robust way to compare two or more independent samples of different sizes. From this test we see that the temperature ($p<0.01$) and light intensities ($p<0.01$) of the different season, and the corresponding BVOC emissions ($p<0.01$) are significantly different at a significance level of more than 3 sigma.

*We have rephrased P6L29-31 of the orginal submission as follows:*
*"Average temperatures were highest in summer (~35±5 °C), followed by the monsoon (~30±8 °C), post-monsoon (~21±7 °C) and winter season (~13.5 ±7°C). Peak hourly PAR ranged from 0-1200 µmol $m^{-2}s^{-1}$ in all seasons except the post-monsoon where maximum hourly values remained below 900 µmol $m^{-2}$ $s^{-1}$ on all days of sampling. The Kruskal-Wallis test results revealed that the temperature ($p<0.01$) and light intensities ($p<0.01$) in different seasons, as well as the corresponding BVOC emissions ($p<0.01$) are significantly different at 99 % confidence interval or more."*

**5)** Relative humidity: it is necessary for the reader to see the humidity data along with figure 1, essential when comparing Monsoon with post-Monsoon data.

**Reply: Done.**
In response to the reviewer's suggestion we have added it to the revised Figure 1 (new Figure 2) as shown below:

**Figure 7 of response and new Figure 2.** BVOC emission fluxes along with PAR, temperature, RH. (expressed in nanomols or picomols per leaf area per second). R: Reproductive growth phase V: Vegetative growth phase.

[Figure]

**6)** Seasonality: To describe inter-seasonal variability of BVOC emissions, the authors seem to have used one single tree (n=1). This is not scientifically acceptable. It is unclear how reproducible the experiment is and what is the intra-species variance of BVOC emissions. Either the experiments are performed with more replicates (at least n=3), or the data should be removed from the manuscript.

**Reply:** We have already addressed this concern in detail and also revised the submission to address these concerns while replying to point 1, which was along similar lines. May kindly refer to the same.

**7)** Seasonality: to describe the seasonal change of BVOC emissions, the authors have performed the measurements during summer, fall (during and post-monsoon) and winter. Why did they not consider spring? The tree phenology strongly changes in spring, which is known as an important player in changing the seasonality of BVOC emissions (e.g. (Fischbach et al., 2002; Noe et al., 2012; Grote et al., 2014; Vanhatalo et al.,2018).

**Reply:** We appreciate the reviewer's comments and points. However, we would like to point out that in contrast to the temperate and boreal environments where "spring" season is long and represents a distinct transition between winter and summer, in the sub-tropical climate for the species *Swietenia macrophylla.*, this period lasts only for about 2-3 weeks wherein the Mahogany tree sheds its old leaves and grows the new leaves simultaneously. We did not measure during this period because 1) having fallen leaves/ leaf litter inside the chamber, which occurs as old leaves fall off would confound the emissions from leaves with leaf litter emissions. 2) accounting for the leaf dry weight or LAI at a time when some leaves fall and new ones replace it was also considered logistically difficult and not worth the effort. Please find below some pictures from our experiments which exhibit the changing leaf phenology during the different seasons.

**Figure 8 of response .** Pictures from the plant chamber experiments which capture the changing leaf phenology during the different seasons

[Figure]

We note that in summer these leaves were still relatively young. While the reviewer's points are well taken and indeed ideally one should be measuring the full year from atleast three different replicates sequentially with three identical chambers and sensors on all of them, in such experimental work, conditions and resources always limit how much can be done. As appreciated by reviewer 2, we still think our dataset has sufficient novelty and several new findings. Future work can certainly build on these first measurements from *Sweetenia macrophylla* King commonly called Mahogany.

**8)** Seasonality: I find two days of measurements of one unique tree, not representative for describing seasonal emission during summer.

**Reply:** We have already addressed this concern in reply to point 1 as a far as replicates are concerned where we also noted that in the pioneering study by Jardine et al. 2015, there were also sometimes only 4 days of measurements from a single tree.
We appreciate the reviewer's concern about 2 days of data because if the metrological in these two days were not representative of summertime, then the emission data may not be representative. We note that the summertime measurements were performed for 52 hours without any break (1 cycle of measurements scanning from m/z 21 to m/z 210 took 3 minutes, so the primary data in fact consists of ~ 780 measurements) after which we had to stop the PTR-MS measurements due to technical issues related to dirty power. The meteorological data of these 52 hours were comparable to typical summer time conditions (low daytime RH and high temperature and PAR) as can be seen from Figure 9 of the response shown below.
We argue therefore that these 52 hours can be taken to represent normal summertime conditions, especially when several previous BVOC emission studies could rely only on offline sample collections few times a day, but still yielded meaningful data and findings.

**Figure 9 of response.** Meteorological conditions in terms of PAR, Temperature and Relative Humidity (RH) during summer before and after the period of the plant chamber experiments during summer (shown as shaded region in the middle)

[Figure]

In the revised version, we clarify the same at Page 7 after line 1 of the original submission as follows:

*"The summertime measurements were performed only for 52 hours, but comparison of the meteorological data for this period with the meteorological data before and after the sampling period showed that the sampling was carried out under conditions characteristic of the typical summer time conditions (low daytime RH and high temperature and PAR)."*

**9)** Methods: it appears that the authors did not make any background correction using empty cuvette. To correctly calculate the BVOC fluxes, background measurements are necessary for taking into account the chemical noise of the cuvette system. However, this point does not seem to be critical since the emissions go to nearly zero during the night. However, data should be corrected before publication.

**Reply:** We did perform experiments to assess emissions from the empty cuvette as well as characterize the transmission of PAR through its material while setting up the experiment. In these experiments we did not detect any contamination/emission from the cuvette at the reported BVOC channels (m/z 63, m/z 69, m/z 81 and m/z 137). The relevant data is shown in Figure 10 of the response below.

In addition, we did subtract the input air into the branch cuvette and considered this as the background for calculating the fluxes as shown in Equation 1. The latter correction would account for contributions to the observed data due to temporal changes in the scrubber efficiency and the PTR-MS instrumental background at these ion detection channels.

**Figure 10 of response.** Example of experimental results showing the background mixing ratios at m/z 63, m/z 69 and sum of m/z 81 and m/z 137 from the empty cuvette even in the afternoon when the cuvette experiences maximum heating.

[Figure]

**10)** Table1: what are the variability of the data?

Reply: Thank you for once again raising this pertinent point about variability, which we missed out on providing in the original submission. As already noted in reply to point 1 we have revised Table 1 of original submission (which is now Table 2 of the revised submission), to include the variability (1 sigma standard deviation) for the seasonally averaged fluxes, and report the average flux from all four trees for winter. Changes to the revised manuscript have already been detailed in reply to point 1 of the reviewer.

**11)** Table1: In the last column, what do "5" and "10" refer to?

**Reply:** We apologize for the confusion. The last column "5" and "10" are the line numbers that got added while formatting the original submission.
These will no longer appear in the revised MS.

**12)** Table2: why the function is different between "vegetative" and "reproductive" phase? This makes the comparison difficult. And what it the working hypothesis behind the use of these 2 functions? If available, please cite relevant literature. Overall the rationale is not described.

**Reply:** The functions are different because the magnitude of BVOC emissions and their dependence of PAR and temperature differs depending on whether the tree is in the vegetative phase or the reproductive phase. This can be noted from the Figure shown below (Figures 11 of response), which show the dependence of BVOC emission fluxes in both phases with PAR and why distinguishing the two phases is meaningful. In the vegetative phase (summer and winter), there is an exponential dependence of the fluxes with PAR whereas in reproductive phase (the time when the tree is flowering and fruiting), there is a linear dependence.

**Figure 11 of response:** Dependence of BVOC emission fluxes in reproductive and vegetative growth phase with PAR.

[Figure]

We respectfully disagree that the rationale was not described. The relevant text in lines 2- 12 at Page 8 of the original submission which appear two paragraphs before Table 2 was discussed in the text, also included citations to the relevant references and are reproduced below for convenience:

"Further, depending on whether the tree's growth is in the reproductive or vegetative phase, the assimilated carbon can be allocated differently impacting the emitted BVOC flux. For example, one could expect that in the vegetative growth phase, emissions of BVOCs would be lower whereas, in the reproductive phase, when flowering and fruiting occur, due to the important functional roles BVOCs play in attracting pollinators and for plant defence, there would be increased emissions of BVOCs (Peñuelas, 2003). Mahogany is known to bear fruits during the monsoon season (Gullison et al., 1996) and trees emit odorous compounds like terpenes for defence purposes especially against herbivores and abiotic stresses like high-intensity light, temperature. This diversion of the carbon allocation for such purposes can decrease growth by diverting photosynthates from the production of vegetative structures (Herms and Mattson, 1992). Henceforth, the two distinct phases are referred to as the vegetative growth phase when the carbon allocation to BVOC synthesis is low and reproductive growth phase, when the carbon allocation by the tree to synthesize BVOCs is high."

We have now added the additional reference to make this even more clear.
*"Further, depending on whether the tree's growth is in the reproductive or vegetative phase (Huijser & Schmid 2011), [...]"*

*Reference: Huijser, P. and Schmid, M., The control of developmental phase transitions in plants, Development 138, 4117-4129 (2011) doi:10.1242/dev.063511*

In consideration to reviewer 2's suggestion, we have now also renamed the variables for the Reproductive phase modeling fn which was **f(T,PAR)**=a*PAR+b*exp(c*T) to **f(T,PAR)**=a*PAR+c*exp(d*T) for easier comparison with the vegetative phase modeling fn and revised Table 2 of the original submission (now Table 3) is as shown below:

**Table 3. Bivariate fit functions and their coefficients for BVOC flux parameterizations as function of PAR and temperature in both the reproductive and vegetative phases of Mahogany**

Vegetative phase modeling fn:

**f(T,PAR)** = a*exp(b*PAR)+c*exp(d*T)

Reproductive phase modeling fn:

**f(T,PAR)** = a* PAR+c*exp(d*T)

| | a | b | c | d | | a | c | d |
|---|---|---|---|---|---|---|---|---|
| Monoterpenes | 0.14 | 0.003 | 0.27 | 0.10 | Monoterpenes | 0.009 | 0.66 | 0.01 |
| Isoprene | 0.01 | 0.002 | 0.000008 | 0.20 | Isoprene | 0.0001 | 0.003 | 0.05 |
| DMS | 1.89 | 0.00001 | 0.02 | 0.16 | DMS | 0.01 | 5.89 | 0.01 |

**13)** It is unclear the calibration procedure used for VOC quantification. Which molecules have been used for the calibration of the PTRMS? Was the standard mixture passing throughout the whole cuvette and canister system during calibration? If the authors have calculated the compounds-specific sensitivities, why did they sum up m/z 81+137 for monoterpene measurements?

**Reply:** We did include this information briefly on Page 6, Lines 6-16 of the original submission and also provided reference to the previous works from our group where we reported the detailed characterization of the instrument, its long term stability, QA/QC of measurements and calibration protocol (Line 1-2; Page 6):
"The instrument has been previously characterized in detail elsewhere (Sinha et al., 2014; Chandra et al., 2017;Kumar et al., 2018)."

However, appreciating the reviewer's comment, we now present details of the calibration procedure, VOC gas standard and calibration experiments as follows:

*This information has now been added to Section 2.2 as the following additional text:*

*"Compound-specific sensitivities (ncps ppb$^{-1}$) were determined using calibration experiments involving dynamic dilution of a VOC gas standard (Apel–Riemer Environmental, Inc., Colorado, USA; containing thirteen VOCs at circa 500 ppb; details provided in Table S1) on 4 May 2018, 4 October 2018, 14 November 2018 and 22 January 2019. The pre-mixed VOC gas standard (Apel–Riemer Environmental,Inc., Colorado, USA) contained 495 ppb of dimethyl sulphide (detected at m/z 63), 483 ppb of isoprene (detected at m/z 69) and 494 ppb of the monoterpene α-pinene (detected at m/z 137 and m/z 81 after fragmentation). The stated accuracy of the VOC standard was 5% for all these compounds and as stated in the manufacturer's certificate several of the compounds remain stable even beyond the one year period mentioned in the certificate. We also verified the same for DMS, isoprene and alpha-*

*pinene by comparison with newer VOC gas standards for which the certificate was still valid and is a standard practice in our laboratory to keep track of any changed concentrations inside the VOC standard after the expiry date (see for e.g. Table S1 of Sinha et al., 2014). The gas standard was dynamically diluted with VOC free-zero air generated using a Gas Calibration Unit (GCU-s v2.1, Ionimed Analytik, Innsbruck, Austria). The flows of both the standard gas and zero air mass flow controllers were measured independently before and after the calibration experiments using a NIST calibrated flow meter (BIOS Drycal definer 220, Mesa Labs, US). Figure S2 presents data from two calibration experiments conducted on 4 May 2018 and 4 October 2018, that show there was very little drift in sensitivity of the PTR-QMS for the three compounds (DMS < 3.8%; isoprene < 4.1 % and alpha-pinene < 6.1 %) even over a period spanning 5 months. The uncertainties were calculated using the root mean square propagation of individual uncertainties including the instrumental precision error, 5% accuracy error inherent in the VOC gas standard and 2% precision error of the MFCs as explained in Sinha et al. 2014. For offline measurements, the standard deviation associated with the average value obtained for each canister measurement already included the instrumental precision error and mass flow controller precision errors. The procedure for calculation of the uncertainties in mixing ratios and emission fluxes has been outlined in the supplement. Table S2 lists the sensitivity factor, limit of detection, instrumental uncertainty and total measurement uncertainty for isoprene, DMS and sum of monoterpenes. The total measurement uncertainty was found to be less than equal to 13% for isoprene, DMS and sum of monoterpenes also accounting for the instrumental background (determined by sampling VOC free air) at these m/z ratios."*

*New Table S1:*
*Table S1: Details of the VOC gas standard (Apel–Riemer Environmental,Inc., Colorado, USA) used in the calibration experiments*

| *Compound* | *Mixing ratio in VOC standard (ppb);Stated accuracy 5%* |
|---|---|
| *Methanol* | *503* |
| *Acetonitrile* | *491* |
| *Methyl vinyl ketone* | *479* |
| *Methyl ethyl ketone* | *497* |
| *Acetaldehyde* | *490* |
| *Acetone* | *493* |
| *DMS* | *495* |
| *Isoprene* | *483* |
| *Benzene* | *492* |
| *Toluene* | *468* |
| *p-Xylene* | *477* |
| *α-pinene* | *494* |
| *1,2,4-Trimethylbenzene* | *510* |

*New Table S2:*

**Table S2: Sensitivity factor, limit of detection, instrumental uncertainty and overall uncertainty of measured VOCs from calibration experiments conducted on 4 May 2018 and 4 October 2018.**

| Calibration performed date (RH) | VOC | Sensitivity factor (ncps ppb⁻¹) | Limit of detection (ppb)* | Instrumental uncertainty (%) | Overall uncertainty (%) |
|---|---|---|---|---|---|
| 04.05.2018 (40%) | DMS | $10.77 \pm 0.14$ | 0.06 | 6 | 10 |
|  | Isoprene | $7.27 \pm 0.13$ | 0.10 | 6 | 10 |
|  | Monoterpenes | $8.21 \pm 0.13$ | 0.07 | 7 | 12 |
| 04.10.2018 (70%) | DMS | $10.42 \pm 0.21$ | 0.12 | 6 | 13 |
|  | Isoprene | $7.01 \pm 0.07$ | 0.04 | 6 | 13 |
|  | Monoterpenes | $7.67 \pm 0.05$ | 0.07 | 7 | 12 |

*\* The limit of detection is defined as $2\sigma$ of the measured normalized signal while measuring zero air (99.999% purity; Sigma gases, New Delhi) divided by the sensitivity.*

**New Figure S2. Results of calibration experiments performed on 4 May 2018 and 4 October 2018 for DMS, isoprene and α-pinene illustrating the excellent linearity and low drift in sensitivity of the PTR-QMS for these compounds**

[Figure]

Concerning why we used the sum of m/z 81 +m/z 137 for monoterpene, we note that monoterpene is not a single compound but a class of compounds having the molecular formula ($C_{10} H_{16}$) and that m/z 137 and m/z 81 ions are used to collectively detect the sum of monoterpenes using the PTR-MS technique. Examples of monoterpenes include 3-carene, myrcene, α-terpinene, γ-terpinene, β-camphene, α-pinene, β-pinene, α-phellandrene, and limonene. As the fragmentation ratio between m/z 137 and m/z 81 ions arising from the monoterpene is not identical for the different monoterpenes, the sum of m/z 81 and m/z 137 is much better for quantification in such case, provided the m/z 81 and m/z 137 can be uniquely attributed to monoterpenes (this is the case for our dataset, please see reply to related point 14 of reviewer 1 as well)

**14)** M/z 81 originate also from other compounds, in particular, LOX products and sesquiterpenes. Can the authors show the correlation between 81 and 137 to rule out that other

VOC were included as monoterpenes? Alternatively, the quantification should be based using only the parent ion (i.e. m/z 137).

**Reply:** We appreciate the reviewer's query which has helped us strengthen the related discussion in the revised MS. In reply to point 1 we have already addressed this concern and also listed the revisions to the text in Section 2.2 concerning the monoterpene measurements as a new paragraph and also added the following new Figure as Figure S8.

*Figure S8: Correlation between observed m81 and m137 signals from the plant chamber output air for all seasons*

[Figure]

**Technical corrections:** We have numbered in order to refer them better.

**T1)** P5L3: How long were the Teflon tubing between cuvette and sampling?

**Reply:** 32 m. We have revised P5L3 to include this information:
*"The total inlet length from the cuvette exit to the instruments was 32 m and considering the inner diameter of 9.5 mm and flow rate of ~ 30 l min-1, the inlet residence time of air was always less than 10 s for the transfer from the cuvette to the instruments housed inside the facility."*

**T2)** Fig1: "nmol/m2s" should be "nmol m-2 s-1"
**Reply:** Agreed and corrected. Please refer to the reply of Specific comment 5.

**T3)** Fig2: the unit of light intensity is missing:
Fig2: please give slopes and intercepts for the linear regressions in the first raw of subplots:
Fig2 legend: please add here the time period of the cumulative BVOC flux and assimilation:

**Reply:** Suggestions have been incorporated also taking into suggestions made by Reviewer 2. Please find the revised Fig. 2 of original submission (now Figure 3 of the revised version) below. Cumulative BVOC emission flux and $CO_2$ assimilation were calculated for every hour of the day and accumulated from sunrise until that hour (P7L33-34). We have added this description to the Figure caption.

*Figure 2 of original submission and new Figure 3 of revised submission: Cumulative BVOC fluxes versus cumulative $CO_2$ assimilation. Cumulative fluxes were calculated for every hour of the day and is accumulated from sunrise until that hour, (b) 3-D plot showing the correlation of the fluxes with instantaneous chamber temperature and PAR for vegetative growth phase and (c) Modeled versus measured VOC fluxes using parameterization presented in Table 3.*

[Figure]

**T4)** P4L6-10: please remove the number of measurements, since this information is misleading:

**Reply:** We are sorry but it is not clear to us as to why providing the number of measurements is misleading. Perhaps the reviewer is suggesting that these numbers need to be clarified? One measurement comprised of one measurement cycle of all ions measured by the PTR-MS. The number of measurements were stated to emphasize that the data derive from primary measurements obtained at high temporal resolution, harnessing the power of the PTR-QMS. In the summer season the BVOC mixing ratios were acquired in mass scan mode sequentially monitoring mass channels from m/z 21-m/z 210 (except m/z 37 and m/z 38) with a dwell time of 0.2 s for m/z 21 ($H_3^{18}O^+$) and m/z 22 and 1s for all other m/z channels, therefore resulting

in a measurement cycle of duration of about 3 minutes. For all other seasons, the BVOC mixing ratios were acquired in the ion selective mode (60 channels) sequentially monitoring mass channels that showed enhancements in summer along with other PTR-MS instrumental background peaks and mass channels that are usually referred to as biogenic (eg: sesquiterpenes) with a dwell time of 0.2 s for m/z 21 ($H_3^{18}O^+$) and 1s for all other m/z channels, therefore resulting in a measurement cycle of duration slightly less than a minute.

To avoid any confusion and also in response to reviewer 2's suggestion to keep these numbers consistent with the hourly data, we now rephrase P4L6-8 as:

*"While sampling and biogenic VOC emission measurements were performed from four Mahogany trees in winter (details in Table 1), the sampling and biogenic VOC emission measurements for three other seasons were from one of the four trees (namely Tree 1 in Table 1) as follows: 2018 summer from 22-24 May (n=52 hours of measurements), 2018 monsoon (n=200 hours of measurements) from 25 September-4 October, 2018 post-monsoon (n=163 hours of measurements) from 15-22 November, and 2019 winter from 24-29 January (n=120 hours of measurements)."*

**T5)** P4L26: please describe the leafage:

**Reply:** Information has been added. The leaf age varied depending on the season in which the measurements were conducted. They ranged from 2-11 months.

**T6)** P5: "The input air was sampled at regular interval". It is unclear when and how often did the authors sample the inlet air.

**Reply:** We clarify this in the revised text on P5L5 of original submission as follows:

*"The input air which served as the background for flux calculations was sampled for all hours of the day in each season by taking measurements 2-3 times a day in each season at different hours of the day"*

**T7)** P6L29-30 Are these temperatures statistically different?
P6L30-31: Are the light intensities statistically different?

**Reply:** Yes, at more than 99% confidence interval based on the Krustal Wallis test. as already clarified in  the reply to specific comment 4 of reviewer 1.

**T8)** P8L31: "paramterization" should be "parameterization"

**Reply:** Done, thank you for pointing out the typo.

**Anonymous Referee #2**

**General Comments**

This manuscript presents 24 (+6 in SI) days of measurements of monoterpene, isoprene and DMS emissions from Mahogany in India. The measurements were conducted using a PTR-Quad and a dynamic branch chamber. The results were compared with modelled emissions and then globally upscaled. The measurements identify Mahogany as one of the missing sources of DMS in the rainforest.

The presented data is novel and even though isoprene and monoterpene emission measurements are more and more common, DMS emission measurements are rare. Therefore, I see the result as interesting for the scientific community and recommend the paper to be accepted.

However, I would like that the Authors addressed following comments before publication:

**Reply:** We thank the reviewer for his/her encouraging and valuable remarks that the presented data is novel and rare and for recommending that the manuscript may be accepted after addressing the comments raised in this insightful and helpful review.

We have numbered the specific comments to refer to them easily below and address them point-wise below:

**Specific comments:**

**1)** One major flaw in the manuscript is the missing discussion about the challenges when measuring DMS (and isoprene). Even though the authors have cited literature (de Gouw et al., 2006; Jardine et al., 2015) which extensively discuss the problem of acetaldehyde clusters' possible influence on mass 63, there is no evidence in this manuscript that this influence was ruled out. Even though, Jardine et al. (2015) stated that in their setup no influence could be seen, their instrumental settings seemed to have been optimized to suppress waterclusters ($H2O \cdot H3O+H3O+ < 4\%$; E/N>145 -> please see my comment P6 L13). There is a sentence (P6 L10-16) stating that isoprene and DMS can be measured at their respective masses without much fragmentation, which is true. As long as the PTR-MS is frequently calibrated under measurement conditions, fragmentation losses (of e.g. isoprene or DMS) are corrected by the sensitivity (this is just the case if the measured and calibrated compound are identical and the signal is above the limit of detection). However, other compounds fragmenting/clustering on the same mass (e.g. M69, M63) are a major source of uncertainty. And therefore identifying/ruling out a possible influence of acetaldehyde to the DMS signal (M63) is crucial. There is a similar issue with MBO, which fragments to M69.

**Reply:** We completely agree with the esteemed reviewer and regret the omission of the discussion and all the careful steps we took for compound assignment in the previous submission. Similar concerns were raised by reviewer1 as well, and we have combined the concerns of both reviewers on this issue. These have already been addressed in detail while replying to point 2 of reviewer 1 and point 14 of reviewer 1 and to avoid repetition of all the new Figures and text and revisions, we request Reviewer 2 to go through those replies.

**2)** It is not very clear when the offline sampling was used. The only references to the offline sampling are in the methods part and in the SI. I concluded, that all data used in the main manuscript were online. Therefore, I would recommend, moving the description of the offline sampling to the SI.

**Reply:** We regret that these details were not clear in our original submission. Though the information was there in the original submission, it was scattered in different places (e.g. figures in main paper and supplement).

In the revised version as mentioned in the reply to point 1 of reviewer 1, we have made the added a new Table (new Table 1 of revised MS; shown above in reply to point 1 of reviewer 1) which lists the exact dates of the online and offline sampling experiments. We request the editor, reviewers and readers to go through the reply to point 1 of reviewer 1 for details.

**3)** P2 L6-8 (also P3 L20-22): Please rephrase (South, Central and North America -> Americas; atmospheric environments -> environments).

**Reply:** Done. We have rephrased "South, Central and North America" to "Americas" and "atmospheric environments" to "environments".

*P2 L6-8 has been rephrased as: "It is widely grown in the American and Asian environments (> 2.4 million km² collectively)."*
*P3L20-22 is rephrased as "The area under this tree in the American and Asian environments collectively exceeds 2.4 million km² of land area."*

**4)** Refer from using the word 'fluxes' (= bidirectional) when you discuss your measurements. As your setup cannot capture deposition, use the word 'emissions' instead (e.g. P4 L2, P4 L5, P6 L28, P6 L31,...)

**Reply:** Agreed.
We have now replaced "fluxes" by "emission fluxes" in all relevant places in the revised manuscript.
These include: *P2L16, P3L13, P4L2, P4L5, P6L28, P6L31, P7L2, P7L4, P7L8, P7L10, P7L11, P7L25, P8L3, P8L16, P8L21, P8L26, P8L31, P9L3, P9L4, P9L5, P9L6, P9L23, P9L24, P10L21* and also in all new text added in replies to reviewer 1.

**5)** P4 L6-8: The number of measurements sounds impressive; however, it is not clear what those measurements are. Is a measurement the measured 1 s dwell time data point? Is it one cycle through all measured compounds? … If the authors want to state the amount of data at all, I would suggest to state the number of 1 h data points, shown e.g. in Fig. 1.

**Reply:** Our profuse apologies for the confusion. As mentioned in reply to point T4 of reviewer 1, we now state the number of 1 h data points for each season.
But to address the reviewer's query, one measurement comprised of one measurement cycle of all ions measured by the PTR-MS. The number of measurements were stated to emphasize that the data derive from primary measurements obtained at high temporal resolution, harnessing the power of the PTR-QMS. In the summer season the BVOC mixing ratios were acquired in mass scan mode sequentially monitoring mass channels from m/z 21-m/z 210 (except m/z 37 and m/z 38) with a dwell time of 0.2 s for m/z 21 ($H_3^{18}O^+$) and m/z 22 and 1s for all other m/z channels, therefore resulting in a measurement cycle of duration of about 3 minutes. For all other seasons, the BVOC mixing ratios were acquired in the ion selective mode (60 channels) sequentially monitoring mass channels that showed enhancements in summer along with other PTR-MS instrumental background peaks and mass channels that are usually referred to as biogenic (eg: sesquiterpenes) with a dwell time of 0.2 s for m/z 21 ($H_3^{18}O^+$) and 1s for all other m/z channels, therefore resulting in a measurement cycle of duration slightly less than a minute.

To avoid any confusion and also in response to reviewer 2's suggestion to keep these numbers consistent with the hourly data, we now rephrase P4L6-8 as:

*"While sampling and biogenic VOC emission measurements were performed from four Mahogany trees in winter (details in Table 1), the sampling and biogenic VOC emission measurements for three other seasons was from one of the four trees (namely Tree 1 in Table 1) as follows: 2018 summer from 22-24 May (n=52 hours of measurements), 2018 monsoon (n=200 hours of measurements) from 25 September-4 October, 2018 post-monsoon (n=163 hours of measurements) from 15-22 November, and 2019 winter from 24-29 January (n=120 hours of measurements)."*

**6)** P4 L9: Omit outdoor (there is no natural indoor environment for Mahogany, I assume).

**Reply:** Agreed and corrected.
*P4L9 is rephrased as: "…growing in the natural environment…"*

**7)** P4 L29: … *using a series of traps containing steel wool, silica gel and activated charcoal.* If those traps were custom build, please state so, otherwise please add the type and brand (this information can be very helpful for people who want to use a similar setup).

**Reply:** These traps were custom built.
*We have corrected line P4 L29 as "… using a series of custom built traps containing steel wool, silica gel and activated charcoal."*

**8)** P5 L2: … *using a second pump by ensuring to have a small positive pressure inside the chamber*…How was this positive pressure achieved (by regulating the flow with a MFC or does this pump have a flow lower than 30 L min-1) and how large was the flow flushing the 60-65 m inlet line?

**Reply:** We ensured a small positive pressure inside the chamber by keeping the second pump's suction rate (~30 l min$^{-1}$) close to but less than that of pump 1.
*We rephrase the statement in P5L2 as "… using a second suction pump which drew slightly less than 30 l min$^{-1}$ thereby ensuring a small positive pressure inside the chamber …"*

**9)** P5 L8: *This is significantly longer than the steady-state* … this statement is correct, however, after installing the chamber a longer equilibration time is necessary to prevent measurement artefacts of physical stress or small injuries of the branch (caused by the installation of the cuvette).

**Reply:** Indeed.
We rephrase P5L6-8 as:

*"After installation of the cuvette, we allowed the branch to acclimatize overnight before starting the measurements to ensure acclimatization/conditioning of leaves to the flows and chamber. This is significantly longer than the steady-state attainment time of circa 5 minutes recommended by Niinemets et al. (2011) but is necessary to prevent measurement artefacts owing to inadvertent physical stress or small injuries to the branch immediately after installation."*

**10)** P6 L6: …*dwell time of 1 s at each m/z channel.* Which channels were measured (stated are M63, M69, M81, M137, however I assume also instrumental background peaks (e.g. M21,

M25, M32, M45, M39, M87) were measured for quality assessment) and what was the measurement cycle time (or what was the sampling frequency)?

**Reply:** Yes, we did monitor 60 m/z channels in all seasons including all the ones listed by the reviewer. This has already been clarified and details including revision to the manuscript have already been provided in replies to point 1 of reviewer 1 and point 5 of reviewer 2.

**11)** P6 L8: Please state the compounds in the calibration gas, as well as the uncertainty of the calibration gas. Furthermore, please state average sensitivities with standard deviation and limit of detection (concentration and emission) for the main compounds (Isoprene, DMS and the calibrated monoterpene compound).

**Reply:** The relevant section 2.2 has already been majorly revised to include details for all these aspects as detailed in replies to points 2 and 11 and T6 of reviewer 1, who made similar queries. To avoid repetition, we refer to these replies above.

**12)** P6 L9: *The total measurement uncertainty was less than 10% for isoprene and DMS and less than 15% for the sum of monoterpenes* ...This low uncertainty seems for me very optimistic for the used setup, especially after addressing my first comment. If I remember correctly the calibration gas from Appel-Riemer has an uncertainty of 5% (valid for 1 year after filling of the gas bottle), then adding uncertainties for 3 MFCs (1 for the inlet at the chamber, 2 for calibration, I assume), 60 – 65 m of tubing, an extra pump, as well as only 1 calibration per season (up to 20 days before measurements). Could the authors please provide the uncertainty calculations for those numbers? Also, are the same values valid for the offline sampling?

**Reply:** We thank the reviewer for the important suggestion and regret the omission in previous version. These concerns and ones raised by reviewer 1 have already been clarified in detail in previous replies (e.g. reply to point 2 of reviewer 1). The revised submissions (main manuscript and supplement) now include additional data from the calibration experiments showing the low drift in sensitivity as well as step wise calculations of sensitivity, mixing ratios and uncertainties in mixing ratios and fluxes. We refer the reviewer to the revised section 2.2 of the new version and revised supplement for these details to avoid unnecessary repetition in the response file.

**13)** P6 L13: I could not find any statement about the used E/N in Jardine et al. (2015). However, from the stated values in their paper (pdrift= 2.0 mbar, Vdrift= 600 V) it seems to be either 145 Td (if the drift temperature was 50°C, like their heated inlet) or 149 Td (if the drift temperature was the more common 60°C). Therefore, their measurements did not fall under the standard operational conditions between 130 and 135 Td. Could the authors please provide average and maximum $H2O \cdot H3O + H3O+$ ratios during this measurement campaign (i.e. was it comparable with the 4% in Jardine et al., 2015)?

**14)** P6 L17: Also state the used drift voltage.

**Reply:** This has been discussed in detail already in reply to point 2 raised by the reviewer. As mentioned therein, the average $H_2O \cdot H_3O^+$ to $H_3O^+$ ratio was only 4.12 % for the entire dataset which is comparable to the 4% or lower abundance during experiments conducted by Jardine et al., 2015.

We have also rephrased P6L17 as:

*"We, therefore, operated the instrument under standard operating conditions of drift tube pressure of 2.2 mbar, drift voltage of 600 V and temperature of 60 °C which yields a Townsend ratio of 135 Td. It resulted in a steady and very high primary ion count (1.3-2.5 x $10^7$ counts per second (cps) $H_3O^+$) and low water cluster (average abundance < 4.1% of primary ion).*

**15)** P7 L3-4: Please add that this statement is valid for winter, as below it is stated that *photosynthetically fixed carbon* (normally associated with PAR) *may be more important than emissions from storage pools* (normally associated with T).

**Reply:** Corrected.
We have rephrased P7L3-4 as: *"Thus, temperature was a major driver for emissions of all three compounds in the winter season."*

**16)** P7 L20: Remove the space in the hyperlink (…ac.uk\cnhgroup…)

**Reply:** Done.
P7L20 is corrected as: *(http://www.es.lancs.ac.uk/cnhgroup/iso-emissions.pdf).*

**17)** P7 L29: Add the year to the Jardine et al. citation.

**Reply:** Done.
P7 L29 is corrected as: *"…in the Jardine et al. (2015) study…"*

**18)** P8 L24-25: *It is also evident that DMS has a strong dependence on temperature, but not on PAR.* The PAR dependency is very hard to see in Fig 2(b). It seems to me quite difficult to state anything about DMS, as there is a rather high offset at low PAR and T (it seems that temperature has no effect at low PAR), there is a huge decline at high T when PAR is around 500 µmol m-2 s-1.

**Reply:** We are thankful to the reviewer for this comment as well as the suggestion in point 21 below (please see details and reply below) to re-orient the axis. While we did state in the next line that this "dependence of DMS flux on temperature is not always followed" and *"We assume that emission of DMS needs an internal sulfur source or uptake of COS (Yonemura et al., 2005;Geng and Mu, 2006) which could be the limiting factor for the huge decline at high temperature."* The revised Figure 3 made in accordance with reviewer2's suggestion does indeed show more clearly that temperature has no effect at low PAR.
We make note of this in the revised MS by adding the following statement at Line 27 Page 8 of original submission:

*"From Figure 3 b it also appears that the temperature has no effect on the DMS emission flux at low PAR (< 400 µmol m$^{-2}$ s$^{-1}$)."*

**19)** P10 L12: Omit outdoor (see earlier comment).

**Reply:** Corrected.
P10L12 is rephrased as: *"growing in their natural environment in India."*

**20)** P14 Figure 1: It seems the PAR axis label has a different color that the PAR graph, please use the same color.

**Reply:** Corrected.
Please find the revised Figure 1 of original submission (Figure 2 in revised submission) shown in specific comment 5 of anonymous referee #1.

**21)** P15 Figure 2:
o (a) Please provide the slopes for the linear fits
o (b) Sadly, these plots are not very clear. They give an idea of the temperature dependence, however, the PAR dependence cannot be seen. I recommend either turning these 3D plots to have the origin (0-point) at the bottom middle and PAR and T axis going with the same angles left and right (=symmetrical) or changing the plot style altogether. Depending on the changes, please state in the figure caption that the color corresponds to the respective VOC flux. Also to make that more obvious, the color bars could be stretched to cover the whole axis (then maybe one set of axis labels would be enough).
o (c) Here I would recommend changing all y axes labels to Modeled flux (with respective unit) and state the compound as a title (centered above each plot)

**Reply:** Thank you for these useful suggestions. The changes have been incorporated and the revised Figure 2 of original submission (Figure 3 of revised submission) is shown below for easy perusal.

Figure 3(a): Cumulative BVOC emission fluxes versus cumulative $CO_2$ assimilation. Cumulative fluxes were calculated for every hour of the day and accumulated from sunrise until that hour, (b) 3-D plot showing the correlation of the emission fluxes with instantaneous chamber temperature and PAR for vegetative growth phase and (c) Modeled versus measured VOC emission fluxes using parameterization presented in Table 3

[Figure]

**22)** P16 Table 2: I recommend renaming the variables in the *Reproductive phase modeling fn:* **f(T,PAR)**=a*PAR+c*exp(d*T) to make it easier to compare to the vegetative phase modeling fn.

**Reply:** Agreed and corrected. The same has already been detailed in reply to specific comment 12 of reviewer 1.

**23)** P17 Table 3: *(x100 nos./km2)*; Please use 100 (or 102) if x100 is a multiplication, similar as in the second column in this table. Please clarify '*nos.*' Is it a unit? If so, please explain it in the table caption.

**Reply:** Corrected. nos. is given as an abbreviation for numbers. We have corrected Table 3 (Table 4 of revised manuscript) according to the referee comment and the same is shown below.

**Table 4. Distribution of Mahogany in natural forests and in plantations in terms of ground area, tree density, leaf area and calculated annual emission fluxes of monoterpenes, isoprene and DMS.**

| Country | Natural Area[i] ($10^4$ km²) | Plantation Area[ii] (km²) | Tree density[iii] Natural/Plantation ($\times 10^2$ km⁻²) | Leaf area[iv] (km²) | Monoterpenes (Gg/yr) | Isoprene (Mg/yr) | DMS (Mg/yr) |
|---|---|---|---|---|---|---|---|
| **Brazil** | 139.6 | - | 0.014-1.17[b]/- | 1564-10756 | 10-69 | 82-565 | 17-119 |
| **Peru** | 56.5 | - | - | 9042 | 58 | 475 | 100 |
| **Bolivia** | 18.9 | - | 0.1-0.2[c]/- | 1512-3025 | 9.7-19 | 79-159 | 17-33 |
| **Nicaragua** | 5 | - | 0.6/- | 2400 | 15 | 126 | 27 |
| **Mexico** | 3.6 | - | 1.0/- | 2881 | 18 | 151 | 32 |
| **Ecuador** | 3.5 | - | - | 2801 | 18 | 147 | 31 |
| **Colombia** | 2.6 | - | - | 2080 | 13 | 109 | 23 |
| **Guatemala** | 2.8 | - | 0.2-2.0/- | 448-4480 | 2.9-29 | 24-235 | 4.9-49 |
| **Honduras** | 1.7 | - | 2.0/- | 2720 | 17 | 143 | 30 |
| **Venezuela** | 1.2 | - | 1.0[d]/- | 960 | 6.1 | 50 | 11 |
| **Panama** | 1 | - | 0.1/- | 80 | 0.5 | 4.2 | 0.88 |
| **Belize** | 1 | 5.91 | 1.0-2.5/119-288[e] | 825-2061 | 5.3-13 | 43-108 | 9.1-23 |
| **Costa Rica** | 0.3 | - | 0.5-2.5/- | 120-600 | 0.77-3.8 | 6.3-32 | 1.3-6.6 |
| **Indonesia** | - | 1160 | - | 3410 | 22 | 179 | 38 |
| **Fiji** | - | 420 | - | 1235 | 7.9 | 65 | 14 |
| **Philippines** | - | 250 | - | 735 | 4.7 | 39 | 8 |
| **Sri Lanka** | - | 45 | - | 132 | 0.85 | 6.9 | 1.5 |
| **Guadeloupe** | - | 40 | - | 118 | 0.75 | 6.2 | 1.3 |
| **Martinique** | - | 15 | - | 44 | 0.28 | 2.3 | 0.49 |
| **Puerto Rico** | - | 13.81 | -/66.7-200[e] | 33-99 | 0.21-0.64 | 1.8-5.2 | 0.37-1.1 |
| **Kerala, India** | - | 1.70[a] | - | 5 | 0.03 | 0.26 | 0.06 |
| **Honduras** | - | 1.50 | - | 4 | 0.03 | 0.23 | 0.05 |
| **St. Lucia** | - | 1.00 | - | 3 | 0.02 | 0.15 | 0.03 |
| **TOTAL** | 237.7 | 1953.92 | | 33154-49674 | 212-317 | 1740-2607 | 366-548 |

[i], [ii,e]Lugo et al. (2003), [iii]Gillies et al. (1999), [a]Mohandas (2000), [b]Grogan et al. (2008), [c]Gullison et al. (1996), [d]Kammesheidt et al. (2001); Leaf Area Index: 2.94 (Jhou et al., 2017)
Crown radius (m)= 0.139 x diameter (cm) - 2.82 $\times 10^{-4}$ x [diameter (cm)]², r² = 0.97 (Gullison et al., 1996)

**24)** PS2 Figure S1: The offline canister sampling is twice in your schematic (once at the KNF pump2 and once behind it). Normally the tubing is marked by 2 lines (e.g. between MFC and Tedlar bag), however right before the instruments it changes to normal arrows, please use only one style.

**Reply:** Thank you for the comment. We have revised it and the offline canister sampling appears only once now. We have also modified the arrows to use only one style.
The revised Figure S1 is given below.

[Figure]

**Figure S1. Schematic of dynamic branch cuvette setup. Offline canister collection scheme is depicted in the dashed rectangle. MFC: Mass flow controller. PTR-MS: proton transfer reaction mass spectrometry. CRDS: Cavity ring down spectroscopy. PAR: Photosynthetically active radiation.**

**25)** PS3 & 5, Figure S3 & S5: The figure caption seems to be wrong, as it says that BVOC emissions are shown, but the y axis unit is ppb. Assumingly it should state Time series of BVOC concentrations in the chamber with corresponding… (in case there are actually emissions shown, please calculate background emissions and change the label).

**Reply:** Sorry for the errors in the Figure captions. We thank the reviewer for his/her careful reading and spotting the error in the caption!

There was no Fig S5 in the original supplement but we think the reviewer meant Fig. S2 and S4.
The Figures S2 and S4 now appear as Figures S9 and S11 and showed the mixing ratio of the gases from the cuvette output air and the background mixing ratios in the cuvette input air
We have corrected the captions as follows in the revised supplement:

*"Figure S9: Time series of BVOC mixing ratios with the corresponding background mixing ratios in nmol mol$^{-1}$ . Background mixing ratios are shown as dotted line."*

and,

*"Figure S11: Time series of winter time BVOC mixing ratios with the corresponding background mixing ratios in nmol mol$^{-1}$ Background mixing ratios are shown as dotted line. Blue shaded region marks a rain event."*

**26)** PS4 & 5, Figure S4 & S5: As there is no reference in the main manuscript to these figures, could you please provide some context to the figures.

**Reply:** We thank the reviewer for noting this. In the revised version, we have now added reference in the main manuscript to this and other supplementary Figures and Tables.

The following line has been added in the main manuscript at P7L13 of original submission to refer these figures.

*"The time series of BVOC mixing ratios in output air of the cuvette alongwith the background mixing ratios in input air are shown in Figure S9 for Tree 1 and Figure S11 for Trees 2,3 and 4, Figure S10 shows the wintertime BVOC emission fluxes for Trees 2,3 and 4 along with PAR and temperature (expressed in nanomols or picomols per m$^2$ leaf area per second)."*

**Technical corrections:**
(see:https://www.atmosphericchemistryandphysics.net/for_authors/manuscript.preparation.html)
**1)**Use SI units (e.g. P4 L19: change to metric) [*For units of **physical quantities**, the metric system is mandatory and, wherever possible, SI units should be used.*]

**Reply:** Corrected.
P4L19 has been re-written as: *"...Dimension: 0.61 m × 0.91 m, 0.05 mm thickness."*

**2)**State the inner diameters of tubing (instead of outer diameters), as those are the crucial parameters for volume, residence time, line losses.

**Reply:** Corrected.
P4L22 is rephrased as: *The bag has one open end and two Jaco fittings (6.3 mm) for inlet and outlet air flow Teflon tubing (0.63 mm, 3.2 mm, 6.3 mm and 9.5 mm I. D., 60-65 m in total with > 95 % length made of 9.5 mm I.D.).*

**3)**Follow the recommendations of the journal for the format of your units (e.g. P2 L13&14, P5 L1,…)
[*Regarding the **notation**, if units of physical quantities are in the denominator, contain numbers, and are abbreviated, they must be formatted with negative exponents (e.g. 10 km h-1 instead of 10 km/h)*]

**Reply:** Done.
We have corrected P2L13-14 as: *"(average in post monsoon: ~19 ng g$^{-1}$ (leaf dry weight) hr$^{-1}$) relative to previous known tree DMS emissions, high monoterpenes (average in monsoon: ~15 µg g$^{-1}$ (leaf dry weight) hr$^{-1}$."*

P5L1 is corrected as: *"at 30 l min⁻¹"*, P5L15 is corrected as: *"of 500 ml min⁻¹"*, P6L3 is corrected as: *"kcal mol⁻¹"*, P6L6 as *"ncps ppb⁻¹"*, P10L14-17 as: *"(average in post monsoon: ~19 ng g⁻¹ (leaf dry weight) hr⁻¹) relative to previous known tree DMS emissions, high monoterpenes (average in monsoon: ~15 µg g⁻¹ (leaf dry weight) hr⁻¹ which are comparable to high emitters such as oak trees) and low emissions of isoprene (< 0.09 µg g⁻¹ (leaf dry weight) hr⁻¹)."*

**4)** Unify the way you state instruments (often type, company, country are stated, sometimes not; e.g. P4 L13, P4 L25, P5 L1, P5 L5,…)

**Reply:** Done.
We have unified the format as: *(Type (model), company, country; other details if any).*
*P4L25 is corrected as: "(VP-4 RH and T sensor, QSO-S PAR sensor, and GS1 SM sensor, Decagon devices, USA)",*
*P5L1 is corrected as: "(EL-FLOW, Bronkhorst High-Tech Netherlands; stated uncertainty 2%)", P5L5 is corrected as: "(BIOS Drycal definer 220, Mesa Labs, US)".*

**5)** Please use either L or l for the unit of liter (e.g. P5 L1, P5 L15,…)

**Reply:** The inconsistency is regretted. We have used l for abbreviating litre.

**6)** Sometimes spaces are missing between values and units (e.g. P4 L15,P9 L21,… )

**Reply:** We have corrected them accordingly.
*P5L15 is corrected as: "250 m", P9L21 is corrected as: "80 cm".*

[revised manuscript text omitted]

The instrument was calibrated atleast four times during the period of study on 4 May, 4 October, 14 November 2018 and 22 January 2019 at different humidities (~ 40 % RH, 60 % RH and 70% RH) using a VOC standard (Apel-Riemer Environmental, Inc., Colorado, USA) by dynamic dilution with zero air at four different mixing ratios (in the range of 2–20 ppbv) for each

VOC. The measured m/z ion signals in counts per second (cps) $(I_{R_iH^+})$ for each VOC was converted to normalized cps (ncps) with respect to sum of reagent $H_3O^+$ ion signal $(I_{H_3O^+})$ and first water cluster $H_3O^+(H_2O)$ signal $(I_{H_3O^+(H2O)})$ using the following normalization equation:

$$ncps = \frac{I_{R_iH^+} \times 10^6}{I_{H_3O^+} + I_{H_3O^+(H2O)}} \times \frac{2}{p_{drift}} \times \frac{T_{drift}}{298.15}$$

The normalized counts per second (ncps) thus calculated was corrected for dilution using zero air using the equation (2):

$$ncps \times Total\ flow = (ncps_{zero} \times Zero\ air\ flow) + (ncps_g \times Standard\ gas\ flow) \quad (1)$$

$$ncps_g = \frac{(ncps \times Total\ flow) - (ncps_{zero} \times Zero\ air\ flow)}{Standard\ gas\ flow} \quad (2)$$

These ncps corrected for dilution (ncps$_g$) were converted to sensitivity (ncps/ppb) by plotting it in y-axis with the introduced concentration of gas standard of each VOC in x-axis. The slope of the graph yielded the sensitivity factor for each VOC which was then used to calculate the mixing ratio (in ppb) from the measured counts per second of each VOC. The standard deviation in ncps$_g$ along with the error in the flows during calibration gives the uncertainty of each VOC measurement. The percentage instrumental uncertainty was then calculated using the root mean square propagation of individual uncertainties like the 5% accuracy error inherent in the VOC gas standard concentration, the $2\sigma$ instrumental precision error while sampling 10 ppbv of the VOC and error in the flow reproducibility (2%) of the two mass flow controllers.

The overall uncertainty in fluxes was calculated by propagating the error in each term in the flux calculation formula and the drift in sensitivity:

$$EF = \frac{m_{out} - m_{in}}{V_m} \times \frac{Q}{A} \quad (1)$$

where, EF is the emission flux, $m_{out} - m_{in}$ is the difference in mixing ratios of the BVOC between input and output air, Q is the flow rate of air passing through the cuvette system in m$^3$ s$^{-1}$, $V_m$ is the molar volume of gas calculated using the cuvette temperature and ideal gas law.

Following are the major steps in calculating the overall uncertainty of fluxes:

Step 1: Let the error in measurement of $m_{out}$ and $m_{in}$ be $s_{out}$ and $s_{in}$ respectively. Since the percentage uncertainties associated with measurement of $m_{out}$ and $m_{in}$ are equal, we can say that $\frac{s_{out}}{m_{out}} = \frac{s_{in}}{m_{in}}$.

Step 2. Uncertainty in measurement of BVOC of difference of input and output air from cuvette.

Let, $y = m_{out} - m_{in}$, $\qquad s_y = \sqrt{s_{out}^2 + s_{in}^2}$ $\qquad (2)$

30    Since we have percentage uncertainties instead of individual absolute uncertainties, $s_y$ can be written as:

$$s_y = \sqrt{m_{out}^2 \left(\frac{s_{out}}{m_{out}}\right)^2 + m_{in}^2 \left(\frac{s_{in}}{m_{in}}\right)^2} = \sqrt{(m_{out}^2 + m_{in}^2)\left(\frac{s_{out}}{m_{out}}\right)^2} = \sqrt{(m_{out}^2 + m_{in}^2)}\frac{s_{out}}{m_{out}} \quad (3)$$

Further simplifying equation (3) we obtain that the maximum relative uncertainty (if $m_{out} = m_{in}$) as:

Therefore the maximum uncertainty ( if $m_{out} = m_{in}$) is given as:

$$s_y = \sqrt{2}\, m_{out} \tag{4}$$

$$\frac{s_y}{y} = \sqrt{2}\, \frac{s_{out}}{m_{out}} \tag{5}$$

5   In the case of plant chamber experiments, $m_{out} \gg m_{in}$ , therefore the maximum uncertainty in difference (y) is 1.4 times the instrumental uncertainty, $\frac{s_{out}}{m_{out}}$.

Step 3: Now since the equation (1) contains only products and quotients to calculate the propagation of error,

$$\frac{s_{EF}}{EF} = \sqrt{\left(\frac{s_y}{y}\right)^2 + \left(\frac{s_Q}{Q}\right)^2 + \left(\frac{s_{V_m}}{V_m}\right)^2 + \left(\frac{s_A}{A}\right)^2 + \left(\frac{s_D}{D}\right)^2} \tag{6}$$

We substitute Eq. (5) with Eq. (4) and by the propagation of individual uncertainties like 2% error in the flow measurement of
10  MFC: (EL-FLOW; Bronkhorst High-Tech), 1.67% error in the leaf area measurement (Easy Leaf Area doi: 10.3732/apps.1400033), uncertainty of molar volume calculation: <1 % (molar volume is calculated theoretically using ideal gas law) and percentage drift in sensitivity (d).

$$\frac{s_{EF}}{EF}(\%) = \sqrt{\left(1.4 \times \text{instrumental uncertainty }(\%)\right)^2 + 2^2 + 1 + 1.67^2 + d^{\,2}} \tag{7}$$

For example, to calculate the total measurement uncertainty (%) in emission fluxes of DMS, isoprene and sum of monoterpenes during post monsoon, we substitute the instrumental uncertainty in mixing ratio and percentage drift in sensitivity of PTR-MS for these 3 compounds (DMS < 3.8%; isoprene < 4.1 % and alpha-pinene < 6.1 %) obtained from calibration experiments
20  conducted on 4 May 2018 and 4 October 2018 in Eq. (7).

$$DMS, \frac{s_{EF}}{EF}(\%) = \sqrt{8.9^2 + 4 + 1 + 1.67^2 + 3.8^{\wedge}2} = 13\ \%$$

$$Isoprene, \frac{s_{EF}}{EF}(\%) = \sqrt{8.9^2 + 4 + 1 + 1.67^2 + 4.1^2} = 13\ \%$$

$$Monoterpenes, \frac{s_{EF}}{EF}(\%) = \sqrt{9.9^2 + 4 + 1 + 1.67^2 + 6.1^2} = 12\ \%$$

The total uncertainty in emissions flux measurements, while not being able to correct between 4 May and 4 October (which spans over 5 months including monsoon season) with new sensitivity, is less than equal to 13% for all the reported VOCs. Thus the calculated total measurement uncertainty can be considered as the upper limit for monsoon season as well.

[Figure]

**Figure S3: A typical 30 min averaged PTR-MS mass scan of the output air from the branch cuvette system during the afternoon period after subtraction of input background air signals showing the ion signals observed in the mass range m/z 40 to m/z 210.**

[Figure]

**Figure S4: Correlation of m/63 ion signal with m/z 45 ion signal for all seasons**

Below we show the chromatographic peak for isoprene from the output air of the branch cuvette system identified based on the retention time of isoprene vapours that were sampled under identical operating conditions with the TD-GC-FID. The isoprene data co-measured with the TD-GC-FID for the monsoon season alongwith the isoprene data measured with the PTR-QMS, was found to have excellent correlation of the PTR-MS isoprene signal but the absolute values were much lower due to the suspected losses within the TD-GC-FID system.

**Isoprene measurements by Thermal Desorption-Gas Chromatography-Flame Ionization Detector (TD-GC-FID):**

Isoprene was detected in output air from the branch cuvette using a gas chromatograph equipped with a flame ionization detector (GC-FID 7890B, Agilent Technologies, Santa Clara, United States) which is coupled to a thermal desorption unit (CIA Advantage-HL and Unity 2, Markes International, UK) for sampling and pre-concentration. Water in the sample air was removed using a nafion dryer which also removed the oxygenated VOCs such as alcohols, aldehydes and ketones (Badol et al., 2004; Gros et al., 2011). 1000 ml of dry sample air was then pre-concentrated at -30°C at 20 ml min$^{-1}$ on an ozone precursor trap (U-T17O3P-2S, Markes Internatioal, UK) which was then thermally desorbed by rapid heating to 325°C. The desorbed analytes were then transferred onto the GC instrument via a heated inlet (130°C) line. The GC instrument consisted of a capillary column (Alumina PLOT, Al$_2$O$_3$/Na$_2$SO$_4$, 50 m x 0.32 mm, 8 μm film thickness, Agilent Technologies, Santa Clara, United States). The oven temperature was ramped from 30°C (hold for 12 min) to 200°C at the rates of 5°C min$^{-1}$ (upto 170°C) and 15°C min$^{-1}$ (upto 200°C) for resolving the peaks.

Isoprene was resolved on Alumina PLOT column at a retention time of 37.5 min and identified based on the retention time of isoprene vapours injected into the TD-GC-FID system under identical instrument operational conditions as the sample. The eluted isoprene was then detected using the FID. Unfortunately due to the suspected transfer losses within the GC system, which could not be corrected, the data is only semi-quantitative and hence reported in arbitrary units.

[Figure]

**Figure S5: Sample chromatogram of the isoprene peak resolved on the Alumina PLOT column at a retention time of 37.5 min in the air collected from the plant chamber experiment overlayed with the peak from pure isoprene vapours injected to determine the retention time of isoprene.**

[Figure]

**Figure S6. Times series of hourly averaged isoprene measurements from PTR-MS and TD-GC-FID for monsoon season**

[Figure]

**Figure S7 Correlation of isoprene data measured with PTR-MS and TD-GC-FID for monsoon season**

[Figure]

**Figure S8: Correlation between observed m81 and m137 signals from the plant chamber output air for all seasons**

[Figure]

Figure S9: Time series of BVOC mixing ratios (hourly averages) with the corresponding background mixing ratios in nmol mol[-1] . Background mixing ratios are shown as dotted line

[Figure]

**Figure S10: Wintertime BVOC emission fluxes along with PAR and temperature. (expressed in nanomols or picomols per leaf area per second). Blue shaded region marks rain event.**

[Figure]

**Figure S11: Time series of winter time BVOC mixing ratios observed for Trees 2, 3 and 4 with the corresponding background mixing ratios in nmol mol$^{-1}$ Background mixing ratios are shown as dotted line. Blue shaded region marks a rain event**

5  **Table S1: Details of the VOC gas standard (Apel–Riemer Environmental,Inc., Colorado, USA) used in the calibration experiments**

| Compound | Mixing ratio in VOC standard (ppb);Stated accuracy 5% |
|---|---|
| Methanol | 503 |
| Acetonitrile | 491 |
| Methyl vinyl ketone | 479 |
| Methyl ethyl ketone | 497 |
| Acetaldehyde | 490 |
| Acetone | 493 |
| DMS | 495 |
| Isoprene | 483 |

| | | |
|---|---|---|
| Benzene | 492 | |
| Toluene | 468 | |
| p-Xylene | 477 | |
| α-pinene | 494 | |
| 1,2,4-Trimethylbenzene | 510 | 5 |

**Table S2. Sensitivity factor, limit of detection, instrumental uncertainty and overall uncertainty of measured VOCs from calibration experiments conducted on 4 May 2018 and 4 October 2018.**

| Calibration performed date (RH) | VOC | Sensitivity factor (ncps ppb$^{-1}$) | Limit of detection (ppb)* | Instrumental uncertainty (%) | Overall uncertainty (%) |
|---|---|---|---|---|---|
| 04.05.2018 (40%) | DMS | $10.77 \pm 0.14$ | 0.06 | 6 | 10 |
| | Isoprene | $7.27 \pm 0.13$ | 0.10 | 6 | 10 |
| | Monoterpenes | $8.21 \pm 0.13$ | 0.07 | 7 | 12 |
| 04.10.2018 (70%) | DMS | $10.42 \pm 0.21$ | 0.12 | 6 | 13 |
| | Isoprene | $7.01 \pm 0.07$ | 0.04 | 6 | 13 |
| | Monoterpenes | $7.67 \pm 0.05$ | 0.07 | 7 | 12 |

* The limit of detection is defined as $2\sigma$ of the measured normalized signal while measuring zero air (99.999% purity; Sigma gases, New Delhi) divided by the sensitivity.

**Table S3. Leaf area and leaf dry weight inside the cuvette during all the experiments**

| Season | Leaf area ($m^2$) | Leaf dry weight (g) | $g/m^2$ |
|---|---|---|---|
| Summer | 0.3 | 30.1 | 96.1 |
| Monsoon | 0.3 | 28.2 | 82.8 |
| Post-monsoon | 0.2 | 20.5 | 109.4 |
| Winter | 0.2 | 26.8 | 135.3 |
| Winter (2) | 0.3 | 27.3 | 102.3 |
| Winter (3) | 0.2 | 31.7 | 138.9 |
| Winter-Offline | 0.3 | 36.1 | 139.8 |

---

## Author Response (AR2)

**High DMS and monoterpene emitting big leaf Mahogany trees: discovery of a missing DMS source to the atmospheric environment**

We thank the anonymous reviewer for his/her encouraging remarks and suggestions. Please find the point-wise revisions/replies (in blue) to the specific points (in black) below.

**Report #1**

The revised version of the manuscript includes several improvements. Overall, the authors have addressed almost all of my previous concerns in a pertinent manner. I have found that the quality of the manuscript has improved considerably. I agree that the manuscript contains a number of important new insights for ACP readers. However, I still have some points that need to be taken into account before publication:

**Reply:** We thank the anonymous reviewer for the critical review and constructive suggestions which helped us improve the manuscript and are grateful for her/his careful reading of the manuscript and deeming that the manuscript can be accepted for publication in ACP subject to addressing the remaining minor comments.

1) VOC identification: monoterpene and isoprene identifications are convincing, unfortunately not yet for the DMS. All the arguments presented by the authors are technically valid. However, the DMS has not been sufficiently validated. Since the main objective of the article is the "discovery of a missing DMS source in the atmospheric environment from mahogany trees", I consider extremely important to provide additional data on the identification of the DMS. From the mass scan data, it is unclear the isotopic distributions of DMS (peaks are too small). What is the percentage of m/z 64 and 65 with respect to 63? Are those peaks reflecting the natural isotopic distribution of 13C, 33S, and 34S? Can you please clearly show it? Another and more robust way to validate DMS is the use of a pure DMS standard and gas chromatography, especially when coupled with mass spectrometry. Honestly, I don't understand why the authors didn't validate the DMS with at least TD-GC-FID in a similar way as they did with isoprene. In fact, the authors possess a standard VOC mixture of Apel-Riemel containing DMS and have (or have access to) a GC-FID instrument. I recommend showing in the supplementary, statistically meaningful GC data (either coupled with FID or and better with MS) of cuvetteenclosed Mahogany trees, compared to background measurements (empty cuvette) and to pure DMS standard measurements.

**Reply:** We appreciate the reviewer for his/her critical comments about the chemical characterisation of DMS and for suggesting that additional data be provided on the identification of DMS. Accordingly, we have followed up on the reviewer's good suggestion and are glad to add the following information at the end of Line 32 Page 8 of the previous submission:

"The measured DMS signals were generally too low to clearly observe the shoulder isotopic peaks originating from the abundance of the 13C, 33S and 34S isotopes. However, during the summer time, when the PTR-MS was operated in the mass scan mode there were periods wherein the DMS signal (m/z 63) was sufficiently high (~0.5 ppb) to observe the isotopic peaks at m/z 64 and m/z 65 (e.g. during noontime on 22.05.2019). Figure S4 (b) shows the 30-minute averaged mass spectra of m/z 63, 64 and 65 during one such occasion. Based on the natural isotopic distribution of 13C, 33S and 34S, one would expect approximately 3.0 % and 4.5 % signal from m/z 63 to land at m/z 64 and 65, respectively and the data in Fig S4 (b) showing the signals

observed at m/z 64 and m/z 65 are consistent with the same. These peaks were also comparable with the mass spectra obtained while calibrating the PTR-MS at DMS mixing ratios of 0.5 ppb. Hence these additional supporting evidence from the shoulder isotopic peaks in combination with previous reports in the literature concerning detection of DMS with PTR-MS provide clear evidence that the signal at m/z 63 observed with the PTR-MS in our dataset can be attributed majorly to DMS."

**Figure 1.** 30 min averaged PTR-MS mass scan of the output air from the branch cuvette system during the afternoon period on 22.05.2018 of m/z 63, 64 and 65

Concerning detection of DMS using GC-MS or TD-GC-FID, we do not have an MS equipped with our GC system but we did try experiments with the TD-GC-FID for DMS identification, which ended in failure as the detection of DMS using a TD-GC-FID system requires specialized traps different from the one in our system. To share our experience freely, our TD-GC-FID sampling and analysis method was configured only for measuring isoprene optimally as that was the biogenic emission we were expecting and not DMS. The TD-GC-FID methodology for isoprene and measurement of other pure hydrocarbons required the use of a NAFION dryer which removes water and polar VOCs such as oxygenated VOCs and DMS. When we did discover the DMS in our PTR-MS data season after season, we considered bypassing the NAFION dryer so as to sample the plant chamber air containing DMS directly but we were faced with the following additional problems:

- Doing a run with the Apel riemer standard which is a mixture containing DMS and other compounds such as acetaldehyde and acetone, did not give clear results as the peak for DMS could not be distinguished from the peaks and signal of the other oxygenated VOCs such as acetaldehyde and acetone.
- 2) The sensitivity of the FID is the lowest to DMS as the FID signal response scales directly with the number of carbon atoms in the analyte molecule. The plant chamber air's high water vapour content would have spoilt/complicated the separation of compounds on the Alumina column and coupled with the low DMS sensitivity in the FID detector, therefore the feasibility of DMS detection in the chamber air was doubtful. While we appreciate the reviewer's point that TD-GC-FID systems can be used for DMS detection, the methodology is not as simple as that for isoprene detection in our system for which all components and methods were optimized. For DMS instead it requires the use of specially designed traps that remove water and other potentially interfering compounds selectively but not DMS before elution through the column, and unfortunately we do not have such a system yet.
- 3) Finally, as already reported in the previous response file during the interactive discussion, during tests of the TD-GC-FID system, variable transfer losses were suspected to have occurred in the system likely within the pre-concentration unit or during transfer from the trap to the column within the TD-GC-FID system. Thus, considering all the above three problems, we could not perform the experiments

Thus, considering all the above three problems, we could not perform the experiments targeted at detection of DMS using TD-GC-FID successfully.

Nonetheless we consider that the isotopic abundance plot provides sufficient evidence to demonstrate that the m63 detected using the PTR-MS is majorly attributable to DMS and so we deem that we have adequately addressed this concern of the reviewer and are grateful to the reviewer for the kind suggestion to utilize the shoulder isotopic peaks.

2) The DMS emissions from Mahogany trees are reported as "high" in the title. The summer emissions rates calculated are in the range of 19.2 ng g-1 hr-1 (+/-19 sd, Table2), or max ~15 pmol m-2 s-1 (Figure1). Despite the large uncertainty, the average emission rate is thousandfold lower than what is known for a strong VOC emitter. For comparison, the isoprene emission capacity of oaks and poplars are in the range of 20-80 nmol m-2 s-1. Even when compared to VOCs emitted by the same plant species (Figure1), DMS emissions are the lowest, i.e. 100 folds lower than monoterpenes and 10 folds lower than isoprene. From the text, I understand that the "high" is just relative to general plant DMS emissions. However, the use of "high" in the text should be used with care, but it is inappropriate in the title and therefore should be removed. It would be more appropriate to say something like this: "Big-leaf Mahogany trees are significant sources of DMS and monoterpene emitted into the atmosphere".

**Reply:** Agreed. We have modified the title in response to the reviewer's suggestion to: "Significant emissions of DMS and monoterpenes by big leaf Mahogany trees: discovery of a missing DMS source to the atmospheric environment"

3) I appreciate the efforts to include the means of biological replicates and the variability of the emissions rates. I understand the willingness to use the data collected form Tree 1 to roughly show seasonal changes of VOCs. However, it is unclear how a seasonal study of VOC can be based on a unique individual of a population and how this should be representative and statistically meaningful. Even when the authors use the data collected in winter to compare Tree 1 to the mean value of Tree 2-4 (Fig.1), there is no evidence on the emission variability

among Tree 1 and the other trees in other seasons. Because seasonal changes of VOCs might be strongly plant-specific (in particular those controlled enzymatically), the relationship between Tree 1 and Tress 2-4 seen in winter might not hold e.g., in summer. The uncertainties on seasonal emissions derived from the biological plant-to-plant variability should be better acknowledged. As a remark, I don't agree to publish works without an appropriate number of replicates, even when poor and scientifically unacceptable studies conducted without adequate repetitions can be found (unfortunately) in literature.

**Reply:** We thank the reviewer for this point which is indeed important to acknowledge and highlight as a limitation of the present study. To make this point precisely and more clearly we have added new text (shown in italics below) in the Conclusions section on Page 10 of the original submission as follows:

"We acknowledge, however, that data from more replicates would be better to characterize the intra-species variability and should be addressed in future studies *and the reported seasonal* values in this study need to be treated with caution as seasonal changes of VOCs could be strongly tree-specific especially when the emissions are controlled by enzymatic processes."

4) In statistics, "n" refers to the sample size or elements in a sample. In P4L6-8 "n" is not clear if the "n" refers to the elements in a sample that are used for the correlation study of modelled vs measured fluxes (Fig3).

**Reply**: In the manuscript, "n" refers to the number of hours of measurement in a sample with measurement cycle of a duration of about 3 minutes in summer season and measurement cycle of duration slightly less than a minute during all other seasons. We used the number of hours of measurement as "n" to be consistent since we used hourly averaged data for our analysis.

We clarify the same in the revised MS in Section 2.1, Paragraph 1; by adding the following new text after Line 13 on Page 4 (Section 2.1) of the original submission as follows:

"Here, "n" refers to the number of hours of measurement in a sample with measurement cycle of a duration of about 3 minutes in summer season and measurement cycle of duration slightly less than a minute during all other seasons. We used the number of hours of measurement as "n" to be consistent since we used hourly averaged data for our analysis."

5) In methods, the plant material should be clearly described. Growing conditions, soil propriety, age of the plants, number of leaves enclosed in the bag and their stage of development, position of the leaves in respect to the tree and sun exposition, and any other useful information.

**Reply:** The trees used for this study were growing in silty clay soil in outdoor conditions. Tree 1, 2 and 3 were seven-year-old mahogany trees located near each other whereas tree 4 was five years old and located 250 m away from the prior location. We selected a branch with 30-50 leaves of similar leaf age (ranging from 2-11 months) also ensuring that all the leaves in cuvette received sunlight throughout the day. The cuvette was suspended carefully on the tree branch to minimize the weight stress on the tree and avoid foliage contact within the cuvette.

The above information has been added to the revised MS in Section 2.1, Paragraph 1; at lines 13-15 and 26-28 on Page 4 (Section 2.1) of the original submission as follows:

"Tree 1, 2 and 3 were seven-year-old mahogany trees whereas Tree 4 was five years old. All the trees were growing in silty clay soil in outdoor conditions." And

"Branches with 30-50 leaves of similar leaf age (ranging from 2-11 months) were selected also ensuring that the cuvette received sunlight throughout the day."

6) The statistic paragraph in the method section is missing. In this section, authors should describe the number of biological replicates ("n" as sample size) and "n" as elements in a sample, all statistical tests, levels of significance, and software packages used to perform the statistical analysis. It is important to provide here the justification of the statistical method used in the study.

**Reply: Agreed.**

As desired by the reviewer we have added all this information as a new paragraph in the revised MS to Methods as new Section 2.3 as follows:

[revised manuscript text omitted]

---

## Author Response (AR3)

Dear Professor Rinne,

Greetings and thank you very much for the helpful suggestions concerning the minor revisions. Accordingly, please find our point-wise replies/changes (in black) to the remaining points listed by you (in blue) below.

1) The reviewer is still very critical on the lack of replicates for the seasonal measurements. I also see this as a major shortcoming. However, I would not totally remove the data, but be very careful not to draw too much conclusion from it. Please do the following changes to text to show the indicative nature of these data.

Page 2, lines 21-22: Replace "seasonal patterns" with "indication of seasonal pattern".

Page 3, line 34: Replace "measured seasonal emission fluxes" with "seasonal fluxes suggested by the measurements".

Page 15, line 18: Replace "measured seasonal emission fluxes" with "seasonal fluxes suggested by the measurements".

Page 16, line 4: Replace "seasonal patterns" with "indication of seasonal pattern".

Reply: We thank the editor for the kind and helpful suggestion to address this remaining concern of the reviewer.
We have made changes as specified in all places in the revised MS. A version with track changes is attached which shows compliance in all these places.

2) The usage of "n" is confusing in the manuscript. I suggest using "n_h" (n with subscript h) to indicate that this is number of hours, not e.g. number of biological replicates.
Reply: We thank the editor for the kind and helpful suggestion to address this remaining concern of the reviewer. Accordingly in the revised submission in Section 2.1 Materials and Methods, where "n" was used, we have replaced "n" by "n_h". The same can be noted from the version with tracked changes attached herewith for easy perusal.

We thank you for the kind editorial support!

Vinayak Sinha on behalf of all co-authors

[revised manuscript text omitted]